# Neural ensemble dynamics in dorsal motor cortex during speech in people with paralysis

Sergey D Stavisky[1,2]*, Francis R Willett[1,2], Guy H Wilson[3], Brian A Murphy[4,5], Paymon Rezaii[1], Donald T Avansino[1], William D Memberg[4,5], Jonathan P Miller[5,6], Robert F Kirsch[4,5], Leigh R Hochberg[7,8,9], A Bolu Ajiboye[4,5], Shaul Druckmann[10], Krishna V Shenoy[2,10,11,12,13,14]†, Jaimie M Henderson[1,13,14]†

[1]Department of Neurosurgery, Stanford University, Stanford, United States; [2]Department of Electrical Engineering, Stanford University, Stanford, United States; [3]Neurosciences Program, Stanford University, Stanford, United States; [4]Department of Biomedical Engineering, Case Western Reserve University, Cleveland, United States; [5]FES Center, Rehab R&D Service, Louis Stokes Cleveland Department of Veterans Affairs Medical Center, Cleveland, United States; [6]Department of Neurosurgery, University Hospitals Cleveland Medical Center, Cleveland, United States; [7]VA RR&D Center for Neurorestoration and Neurotechnology, Rehabilitation R&D Service, Providence VA Medical Center, Providence, United States; [8]Center for Neurotechnology and Neurorecovery, Department of Neurology, Massachusetts General Hospital, Harvard Medical School, Boston, United States; [9]School of Engineering and Robert J. & Nandy D. Carney Institute for Brain Science, Brown University, Providence, United States; [10]Department of Neurobiology, Stanford University, Stanford, United States; [11]Department of Bioengineering, Stanford University, Stanford, United States; [12]Howard Hughes Medical Institute, Stanford University, Stanford, United States; [13]Wu Tsai Neurosciences Institute, Stanford University, Stanford, United States; [14]Bio-X Program, Stanford University, Stanford, United States

*For correspondence:
sergey.stavisky@gmail.com

†These authors contributed equally to this work

**Abstract** Speaking is a sensorimotor behavior whose neural basis is difficult to study with single neuron resolution due to the scarcity of human intracortical measurements. We used electrode arrays to record from the motor cortex 'hand knob' in two people with tetraplegia, an area not previously implicated in speech. Neurons modulated during speaking and during non-speaking movements of the tongue, lips, and jaw. This challenges whether the conventional model of a 'motor homunculus' division by major body regions extends to the single-neuron scale. Spoken words and syllables could be decoded from single trials, demonstrating the potential of intracortical recordings for brain-computer interfaces to restore speech. Two neural population dynamics features previously reported for arm movements were also present during speaking: a component that was mostly invariant across initiating different words, followed by rotatory dynamics during speaking. This suggests that common neural dynamical motifs may underlie movement of arm and speech articulators.

## Introduction

Speaking requires coordinating numerous articulator muscles with exquisite timing and precision. Understanding how the sensorimotor system accomplishes this behavioral feat requires studying its

**eLife digest** Speaking involves some of the most precise and coordinated movements humans make. Learning how the brain produces speech could lead to better treatments for speech disorders. But it can be challenging to study. Human speech is unique, limiting what can be learned from animal studies. There also are few opportunities where it would be safe or ethical to take measurements from inside a person's brain while they talk. Most previous studies have recorded brain activity during speech in patients who have had electrodes placed in the brain for epilepsy or Parkinson's disease treatment.

Now, Stavisky et al. show that brain cells that control hand and arm movements are also active during speech. Two patients who had lost the use of their arms and legs but were able to speak participated in the study. The two individuals were already enrolled in a pilot clinical trial of a brain-computer interface to help them control prosthetic devices. As part of this trial, the volunteer participants had two 100-electrode arrays surgically placed in the part of the brain that controls the movement of the arms and hands.

This study made the unexpected discovery that brain cells multitask controlling not just arm and hand movements, but also carry information about movements of the lips, tongue and mouth necessary for speech. Stavisky et al. also found similarities in the patterns of brain activity during hand and arm movements and speech.

By analyzing the activity in these brain cells as the two individuals recited words and syllables, Stavisky et al. were also able to train computers to identify which sound the person spoke from the brain activity alone. This is a first step towards developing a technology that could synthesize speech from a person's brain activity as they try to speak. Much more work is needed to synthesize continuous speech. But the study provides initial evidence that it might be possible to use recordings from inside the brain to one day restore speech to individuals who have lost it.

neural underpinnings, which are critical for identifying (*Tankus and Fried, 2018*) and treating the causes of speech disorders and for building brain-computer interfaces (BCIs) to restore lost speech (*Guenther et al., 2009*; *Herff and Schultz, 2016*). Speaking is also a uniquely human behavior, which presents a high barrier to electrophysiological investigations. Previous direct neural recordings during speaking have come from electrocorticography (ECoG) (*Bouchard and Chang, 2014*; *Cheung et al., 2016*; *Mugler et al., 2014*) or single-unit (SUA) recordings from penetrating electrodes during the course of clinical treatment for epilepsy (*Chan et al., 2014*; *Creutzfeldt et al., 1989*; *Tankus et al., 2012*) or deep brain stimulation for Parkinson's disease (*Lipski et al., 2018*; *Tankus and Fried, 2018*). Such studies have begun to characterize motor cortical population dynamics underlying speech (*Bouchard et al., 2013*; *Chartier et al., 2018*; *Pei et al., 2011*), but not at the finer spatial scale (compared to ECoG) or across larger neural ensembles (compared to single electrodes) afforded by the high-density intracortical recordings widely used in animal studies (*Allen et al., 2019*; *Cohen and Maunsell, 2009*; *Kiani et al., 2014*; *Smith and Kohn, 2008*), including those examining arm reaching (*Carmena et al., 2003*; *Churchland et al., 2012*; *Kaufman et al., 2016*; *Maynard et al., 1999*).

We studied speech production at this resolution by recording from multielectrode arrays previously placed in human motor cortex as part of the BrainGate2 BCI clinical trial (*Hochberg et al., 2006*). This research context dictated two important elements of the present study's design. First, both participants had tetraplegia due to spinal-cord injury but were able to speak; this enabled observing motor cortical spiking activity during overt speaking, in contrast to earlier studies of attempted speech by participants unable to speak (*Brumberg et al., 2011*; *Guenther et al., 2009*). However, these participants' long-term paralysis means that their neurophysiology may differ from that of people who are able-bodied; we will discuss the need for interpretation caution in the Discussion.

Second, the electrode arrays were in dorsal 'hand knob' area of motor cortex, which we previously found to strongly modulate to these participants' attempted movement of their arm and hand (*Ajiboye et al., 2017*; *Brandman et al., 2018*; *Pandarinath et al., 2017*). Speech-related activity has not previously been reported in this cortical area, but there are several hints in the literature that

dorsal motor cortex may have speech-related activity. Although imaging experiments consistently identify ventral cortical activation during speaking tasks, a meta-analysis of such studies (*Guenther, 2016*) indicates that responses are occasionally seen (though not, to our knowledge, explicitly called out) in dorsal motor cortex. Additionally, behavioral (*Gentilucci and Campione, 2011*; *Vainio et al., 2013*), transcranial magnetic stimulation studies (*Devlin and Watkins, 2007*; *Meister et al., 2003*), and electrical stimulation mapping studies (*Breshears et al., 2018*) have reported interactions (and interference) between motor control of the hand and mouth. This close linkage between hand and speech networks has been hypothesized to be due to a need for hand-mouth coordination and an evolutionary relationship between manual and articulatory gestures (*Gentilucci and Stefani, 2012*; *Rizzolatti and Arbib, 1998*). Here, we explicitly set out to test whether neuronal firing rates in this dorsal motor cortical area modulated when participants produced speech and orofacial movements.

## Results

### Speech-related activity in dorsal motor cortex

We recorded neural activity during speaking from participants 'T5' and 'T8', who previously had two arrays each consisting of 96 electrodes placed in the 'hand knob' area of motor cortex (*Figure 1A, B*). The participants performed a task in which on each trial they heard one of 10 different syllables or one of 10 short words, and then spoke the prompted sound after hearing a go cue (*Figure 1—figure supplement 1* shows audio spectrograms and reaction times for these tasks). We analyzed both sortable SUA that could be attributed to an individual neuron's action potentials (*Figure 1C,D*), and 'threshold crossing' spikes (TCs) that might come from one or several neurons (*Figure 1—figure supplement 2*). Firing rates showed robust changes during speaking of syllables (*Figure 1*, *Figure 1—figure supplement 2*, *Video 1*) and words (*Figure 1—figure supplement 3*). Significant modulation was found during speaking at least one syllable (p<0.05 compared to during silence) in 73/104 T5 electrodes' TCs (13/22 SUA) and 47/101 T8 electrodes (12/25 SUA). Active neurons were distributed throughout the area sampled by the arrays, and most modulated to speaking multiple syllables (*Figure 1B* and *Figure 1—figure supplement 4*), suggesting a broadly distributed coding scheme. This is consistent with previous single neuron recordings in the temporal lobe (*Creutzfeldt et al., 1989*; *Tankus et al., 2012*).

Three observations lead us to believe that this neural activity is related to motor cortical control of the speech articulators (*Chartier et al., 2018*; *Conant et al., 2018*; *Mugler et al., 2018*) rather than perception or language. First, modulation was significantly stronger when speaking compared to after hearing the auditory prompts: the neural population firing rate change compared to the silent condition was 4.03 times higher after the go cue compared to after the audio prompt for the T5-syllables dataset, 2.90x for the T8-syllables dataset (*Figure 1E*), 6.71x higher for the T5-words dataset, and 2.12x for T8-words (*Figure 1—figure supplement 3*). Modulation following the audio prompt, although small, was significant when compared to a 1 s epoch just prior to the prompt (p<0.01, sign-rank test, all four datasets). In this study, we are unable to disambiguate whether this prompt-related response reflects auditory perception, movement preparation, or small overt movements preceding vocalization. We will primarily focus on the larger, later neural modulation putatively related to speech production.

Second, analysis of an additional dataset in which participant T5 spoke 41 different phonemes revealed that neural population activity showed phonemic structure (*Figure 1—figure supplement 5*): for example, when phonemes were grouped by place of articulation (*Bouchard et al., 2013*; *Lotte et al., 2015*; *Moses et al., 2019*), population firing rate vectors were significantly more similar between phonemes within the same group than between phonemes in different groups (p<0.001, shuffle test). Third, in both participants, 99 of 120 electrodes that were active during speaking syllables (24 out of 25 sorted neurons) also were active during production of at least one of seven non-speech orofacial movements (*Figure 2* and *Figure 2—figure supplement 1*). We also observed weak but significant firing rate correlations with breathing (*Figure 2—figure supplement 2*). Modulation for speaking was stronger than for unattended breathing (~4.7 x) and instructed breathing (~2.6 x), and modulation for attempted arm movements was stronger than for speaking and orofacial movements (~2.8 x, *Figure 2—figure supplement 3*).

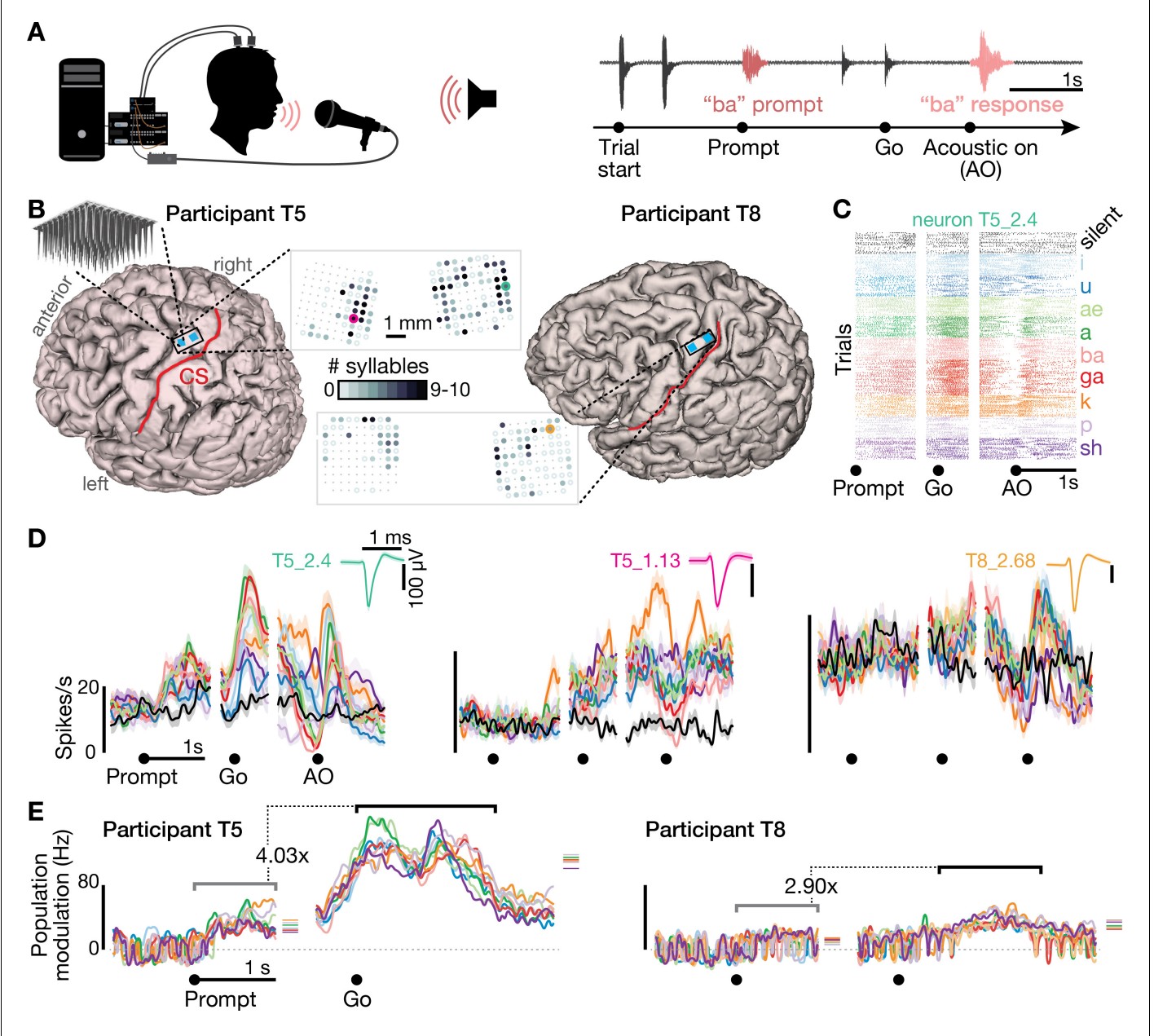

**Figure 1.** Speech-related neuronal spiking activity in dorsal motor cortex. (**A**) Participants heard a syllable or word prompt played from a computer speaker and were instructed to speak it back after hearing a go cue. Motor cortical signals and audio were simultaneously recorded during the task. The timeline shows example audio data recorded during one trial. (**B**) Participants' MRI-derived brain anatomy. Blue squares mark the locations of the two chronic 96-electrode arrays. Insets show electrode locations, with shading indicating the number of different syllables for which that electrode recorded significantly modulated firing rates (darker shading = more syllables). Non-functioning electrodes are shown as smaller dots. CS is central sulcus. (**C**) Raster plot showing spike times of an example neuron across multiple trials of participant T5 speaking nine different syllables, or silence. Data are aligned to the prompt, the go cue, and acoustic onset (AO). (**D**) Trial-averaged firing rates (mean ± s.e.) for the same neuron and two others. Insets show these neurons' action potential waveforms (mean ± s.d.). The electrodes where these neurons were recorded are circled in the panel B insets using colors corresponding to these waveforms. (**E**) Time course of overall neural modulation for each syllable after hearing the prompt (left alignment) and when speaking (right alignment). Population neural distances between the spoken and silent conditions were calculated from TCs using an unbiased measurement of firing rate vector differences (see Methods). This metric yields signed values near zero when population firing rates are essentially the same between conditions. Firing rate changes were significantly greater (p < 0.01, sign-rank test) during speech production (comparison epoch shown by the black window after Go) compared to after hearing the prompt (gray window after Prompt). Each syllable's mean modulation across the comparison epoch is shown with the corresponding color's horizontal tick to the right of the plot. The vertical scale is the same across participants, revealing the larger speech-related modulation in T5's recordings.

*Figure 1 continued on next page*

*Figure 1 continued*

The online version of this article includes the following figure supplement(s) for figure 1:

**Figure supplement 1.** Prompted speaking tasks behavior.
**Figure supplement 2.** Example threshold crossing spike rates.
**Figure supplement 3.** Neural activity while speaking short words.
**Figure supplement 4.** Neural correlates of spoken syllables are not spatially segregated in dorsal motor cortex.
**Figure supplement 5.** Neural activity shows phonetic structure.

## Speech can be decoded from intracortical activity on individual trials

We next performed a decoding analysis to quantify how much information about the spoken syllable or word was present in the time-varying neural activity. Multi-class support vector machines were used to predict the spoken sound (or silence) from single trial TCs and high-frequency LFP power (*Figure 3*). Cross-validated prediction accuracies for syllables were 84.6% for T5 (10 classes, mean chance accuracy was 10.1% across shuffle controls) and 54.7% for T8 (11 classes, chance was 8.6%). Word decoding accuracies were 83.5% for T5 (11 classes, chance was 9.1%) and 61.5% for T8 (11 classes, chance was 9.3%). We also used the same method to decode neural activity from 0 to 500 ms after the speech prompt and found that classification accuracies were only marginally better than chance (overall accuracies between 11.1% and 16.6% across the four datasets, p<0.05 versus shuffle controls in three of the four datasets; *Figure 3C* gray bars). The much higher neural discriminability of syllables and words during speaking rather than after hearing the audio prompt is consistent with the previously enumerated evidence that modulation in this cortical area is related to speech production.

During speaking, decoding accuracies for all individual sounds were above chance (p<0.01, shuffle test). Decoding mistakes (*Figure 3B*) and low-dimensional representations (*Figure 3A*) tended to follow phonetic similarities (e.g. *ba* and *ga*, *a* and *ae*). This observation is consistent with previous ECoG studies (*Bouchard et al., 2013*; *Cheung et al., 2016*; *Livezey et al., 2019*; *Moses et al., 2019*; *Mugler et al., 2014*), although the larger neural differences we observed between unvoiced *k* and *p* and the beginning of their voiced counterparts at the start of *ga* and *ba* suggests strong laryngeal tuning (*Dichter et al., 2018*). These neural correlate similarities may reflect similarities in the underlying articulator movements (*Chartier et al., 2018*; *Lotte et al., 2015*; *Mugler et al., 2018*).

## Neural population dynamics exhibit low-dimensional structure during speech

These multielectrode recordings enabled us to observe motor cortical dynamics during speech at their fundamental spatiotemporal scale: neuron spiking activity. Specifically, we examined whether two known key dynamical features of motor cortex firing rates during arm reaching were also present during speaking. Importantly, both of these features were revealed when looking not at individual neurons' firing rates, but rather were seen when examining the time courses of population activity 'components' that

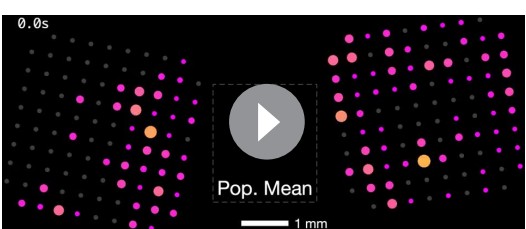

**Video 1.** Example audio and neural data from eleven contiguous trials of the prompted syllables speaking task. The audio track was recorded during the task and digitized alongside the neural data; it starts with the two beeps indicating trial start, after which the syllable prompt was played from computer speakers, followed by the go cue clicks, and finally the participant speaking the syllable. The video shows the concurrent $-4.5 \times$ RMS threshold crossing spikes rate on each electrode. Each circle corresponds to one electrode, with their spatial layout corresponding to electrodes' locations in motor cortex as in the *Figure 1B* inset. Each electrode's moment-by-moment color and size represent its firing rate (soft-normalized with a 10 Hz offset, smoothed with a 50 ms s.d. Gaussian kernel). The color map goes from pink (minimum rate across electrodes) to yellow (maximum rate), while size varies from small (minimum rate) to large (maximum rate). Non-functioning electrodes are shown as small gray dots. To assist the viewer in perceiving the gestalt of the population activity, a larger central disk shows the mean firing rate across all functioning electrodes, without soft-normalization. Data are from the T5-syllables dataset, trial set #23.
https://elifesciences.org/articles/46015#video1

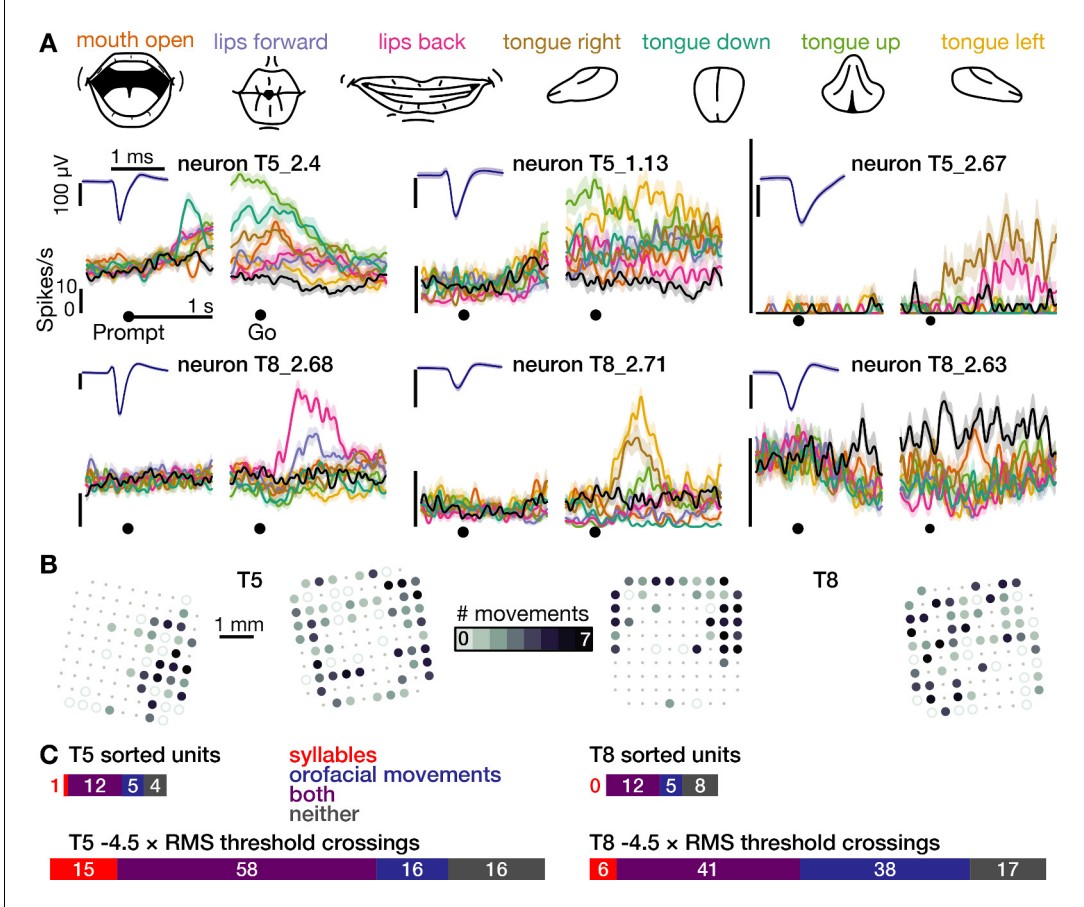

**Figure 2.** The same motor cortical population is also active during non-speaking orofacial movements. (**A**) Both participants performed an orofacial movement task during the same research session as their syllables speaking task. Examples of single neuron firing rates during seven different orofacial movements are plotted in colors corresponding to the movements in the illustrated legend above. The 'stay still' condition is plotted in black. The same three example neurons from **Figure 1D** are included here. The other three neurons were chosen to illustrate a variety of observed response patterns. (**B**) Electrode array maps indicating the number of different orofacial movements for which a given electrode's −4.5 × RMS threshold crossing rates differed significantly from the stay still condition. Data are presented similarly to the **Figure 1B** insets. Firing rates on most functioning electrodes modulated for multiple orofacial movements. See **Figure 2—figure supplement 1** for individual movements' electrode response maps. (**C**) Breakdown of how many neurons' (top) and electrodes' TCs (bottom) exhibited firing rate modulation during speaking syllables only (red), non-speaking orofacial movements only (blue), both behaviors (purple), or neither behavior (gray). A unit or electrode was deemed to modulate during a behavior if its firing rate differed significantly from silence/staying still for at least one syllable/movement.

The online version of this article includes the following figure supplement(s) for figure 2:

**Figure supplement 1.** Neural correlates of orofacial movements are not spatially segregated in dorsal motor cortex.
**Figure supplement 2.** Dorsal motor cortex correlates of breathing.
**Figure supplement 3.** Dorsal motor cortex modulates more strongly during attempted arm and hand movements than orofacial movements and speaking.

act as lower dimensional building blocks (or condensed summaries) of the many individual neurons' activities (**Gallego et al., 2017**; **Pandarinath et al., 2018**; **Saxena and Cunningham, 2019**). The first prominent neural population dynamics feature ('dynamical motif') we tested for is inspired by previous nonhuman primate (NHP) experiments showing that the neural state undergoes a rapid change during movement initiation which is dominated by a condition-invariant signal (CIS) (**Kaufman et al., 2016**). In that study, Kaufman and colleagues provide a comprehensive exposition on why a large neural component that is highly invariant across many different arm reaches is a non-trivial feature of neural population data and could, despite its non-specificity, be important to the overall computations being performed. A similar CIS was recently also reported during NHP grasping (**Intveld et al., 2018**).

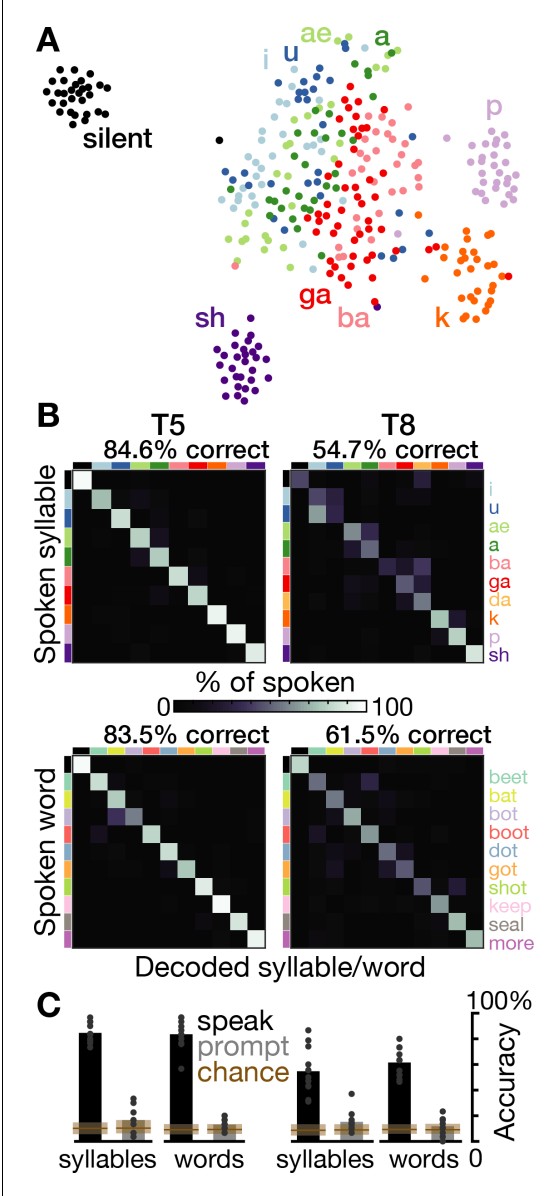

**Figure 3.** Speech can be decoded from intracortical activity. (**A**) To quantify the speech-related information in the neural population activity, we constructed a feature vector for each trial consisting of each electrode's spike count and HLFP power in ten 100 ms bins centered on AO. For visualization, two-dimensional t-SNE projections of this feature vector are shown for all trials of the T5-syllables dataset. Each point corresponds to one trial. Even in this two-dimensional view of the underlying high-dimensional neural data, different syllables' trials are discriminable and phonetically similar sounds' clusters are closer together. (**B**) The high-dimensional neural feature vectors were classified using a multiclass SVM. Confusion matrices are shown for each participant's leave-one-trial-out classification when speaking syllables (top row) and words (bottom row). Each matrix element shows the percentage of trials of the corresponding row's sound that were classified as the sound of the corresponding column. Diagonal elements show correct classifications. (**C**) Bar heights show overall classification accuracies for decoding neural activity during speech (black bars, summarizing panel B) and decoding neural activity following the audio prompt (gray bars). Each small point corresponds to the accuracy for one class (silence, syllable, or word). Brown boxes show the range of chance performance: each box's bottom/center/top correspond to minimum/mean/maximum overall classification accuracy for shuffled labels.

The second dynamical motif we tested for follows studies of NHP arm reaches (*Churchland et al., 2012*; *Kaufman et al., 2016*) and human point-to-point hand movements (*Pandarinath et al., 2015*), which showed that subsequent peri-movement neural ensemble activity is characterized by orderly rotatory dynamics. That is, a substantial portion of moment-by-moment firing rate changes can be explained by a simple rotation of the neural state in a plane that summarizes the correlated activity of groups of neurons. These observations, in concert with neural network modeling (*Kaufman et al., 2016*), have led to a model of motor control in which, prior to movement, inputs specifying the movement goal create attractor dynamics toward an advantageous initial condition (*Shenoy et al., 2013*). During movement initiation, a large transient input 'kicks' the network into a different state from which activity evolves according to rotatory dynamics such that muscle activity is constructed from an oscillatory basis set (akin to composing an arbitrary signal from a Fourier basis set) (*Churchland et al., 2012*; *Sussillo et al., 2015*).

We tested whether motor cortical activity during speaking also exhibits these dynamics by applying the analytical methods of *Churchland et al. (2012)* and *Kaufman et al. (2016)*. These analyses used two different dimensionality reduction techniques (*Cunningham and Yu, 2014*) to reveal latent low-dimensional structure in the trial-averaged firing rates for different conditions (here, speaking different words). Both methods sought to find a modest number of linear weightings of different electrodes' firing rates (forming the aforementioned neural population activity 'components') that capture a large fraction of the overall variance. This is akin to principal components analysis (PCA), but unlike PCA, each method also looks for a specific form of neural population structure: jPCA (*Churchland et al., 2012*) seeks components with rotatory dynamics, whereas dPCA (*Kaufman et al., 2016*; *Kobak et al., 2016*) decomposes neural activity into CI and condition-dependent (CD) components. Importantly, these methods do not spuriously find the sought dynamical structure when it is not present in the data (*Churchland et al., 2012*; *Elsayed and Cunningham, 2017*; *Kaufman et al., 2016*; *Kobak et al., 2016*; *Pandarinath et al., 2015*).

We found that these two prominent population dynamics motifs were indeed also present during speaking. Like in *Kaufman et al. (2016)*, the largest dPCA component summarizing each participants' neural activity during movement initiation was largely CI: this component was 98.7% CI in participant T5, and 87.3% CI in participant T8 (*Figure 4A*). *Figure 4B* shows that in T5, this 'CIS$_1$' component, which rapidly increased after the go cue, was essentially identical regardless of which word was spoken. In T8, the CIS$_1$ was not as cleanly condition-invariant, but nonetheless showed a similar increase following the go cue for each word. We also found this condition-invariant neural population activity component in all four additional datasets that we examined: T5's and T8's syllables task datasets, as well as two additional replication datasets in which

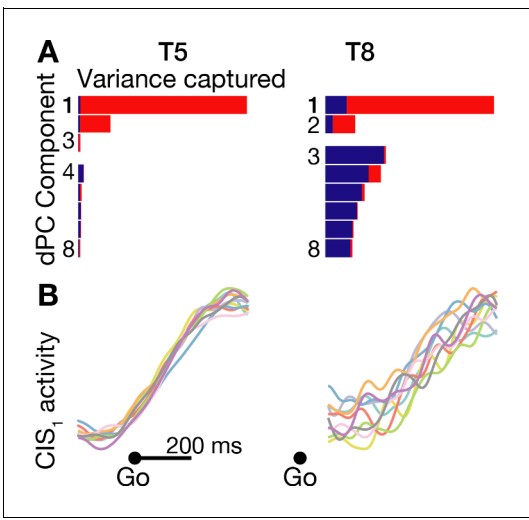

**Figure 4.** A condition-invariant signal during speech initiation. (**A**) A large component of neural population activity during speech initiation is a condition-invariant (CI) neural state change. Firing rates from 200 ms before to 400 ms after the go cue (participant T5) and 100 ms to 700 ms after the go cue (T8) were decomposed into dPCA components like in *Kaufman et al. (2016)*. Each bar shows the relative variance captured by each dPCA component, which consists of both CI variance (red) and condition-dependent (CD) variance (blue). These eight dPCs captured 45.1% (T5-words) and 8.4% (T8-words) of the overall neural variance, which includes non-task related variability ('noise'). (**B**) Neural population activity during speech initiation was projected onto the first dPC dimension; this 'CIS$_1$' is the first component from panel A. Traces show the trial-averaged CIS$_1$ activity when speaking different words, denoted by the same colors as in *Figure 3B*.

The online version of this article includes the following figure supplement(s) for figure 4:

**Figure supplement 1.** Further details of neural population dynamics analyses and additional datasets.
**Figure supplement 2.** Neural population dynamics when viewed across a range of reduced dimensionalities.

participant T5 spoke just five of the words (*Figure 4—figure supplement 1*). These results were also robust across different choices of how many dPCs to summarize the neural population activity with (*Figure 4—figure supplement 2*).

We attribute the difference in how condition-invariant the $CIS_1$ component was between the two participants to the much smaller speech task-related neural modulation recorded in participant T8 compared to in T5, as demonstrated in *Figure 1—figure supplement 3B* and the lower classification accuracies of *Figure 3*. The practical consequence of T8's substantially weaker speech-related modulation is that much more of the neural population activity that dimensionality reduction tries to summarize was not task-relevant (i.e. is 'noise' for the purpose of these analyses). This lower signal-to-noise ratio can also be appreciated in how the 'elbow' of T8's cumulative neural variance explained by PCA or dPCA components (*Figure 4—figure supplement 1A,B*) occurs after fewer components and explains far less overall variance.

Lastly, we looked for rotatory population dynamics around the time of acoustic onset. *Figure 5A* shows ensemble firing rates projected into the top jPCA plane (i.e. the subspace defined by $jPC_1$ and $jPC_2$). In participant T5, all conditions' neural state trajectories rotated in the same direction (similarly to *Churchland et al., 2012*; *Pandarinath et al., 2015*), and rotatory dynamics could explain substantial variance in how population activity evolved moment-by-moment during speaking. Application of a recent population dynamics hypothesis testing method (*Elsayed and Cunningham, 2017*) revealed that this rotatory structure was significantly stronger than expected by chance in T5's speaking data, but not in T8's speaking data (*Figure 5B*) or when this analysis was applied to neural activity following the audio prompt (*Figure 4—figure supplement 1H*). As was the case for the condition-invariant dynamics, these results were also consistent across additional datasets (*Figure 4—figure supplement 1E–H*) and across the choice of how many PCA dimensions in which to look for rotatory dynamics (*Figure 4—figure supplement 2B*). We again attribute the observed between-participants difference to T8's smaller measured neural responses during speech, which likely reflect his older arrays' lower signal quality. Consistent with this, T8's BCI computer cursor control performance was also substantially worse than T5's (*Pandarinath et al., 2017*). Other factors that could also have contributed to T8's reduced speech-related neural activity include his tendency to speak quietly and with less clear enunciation (consistent with *Jiang et al., 2016*), array placement differences, and differences in cortical maps between individuals (*Farrell et al., 2007*).

*Videos 2* and *3* show the temporal relationship between these two dynamical motifs – an initial condition-invariant neural state shift, followed by rotatory dynamics. Neural state rotations occurred after the condition invariant translation; by comparison, in *Kaufman et al. (2016)* the neural rotations also lagged the CIS shift, but in the monkey arm reaching data these rotations either partially overlapped with, or more immediately followed, the CIS shift. We note that existing models of how a condition-invariant signal 'kicks' dynamics into a different state space region where rotatory dynamics unfold (*Kaufman et al., 2016*; *Sussillo et al., 2015*) do not require that the CIS and rotatory dynamics must be orthogonal, but in these data we did observed that the $CIS_1$ and jPCA dimensions were largely orthogonal (*Figure 4—figure supplement 1E*).

## Discussion

There are three main findings from this study. First, these data suggest that 'hand knob' motor cortex, an area not previously known to be active during speaking (*Breshears et al., 2015*; *Dichter et al., 2018*; *Leuthardt et al., 2011*; *Lotte et al., 2015*), may in fact participate, or at least receive correlates of, neural computations underlying speech production. Speech-related single-neuron modulation might have been missed by previous studies due to the coarser resolution of ECoG (*Chan et al., 2014*). If this finding holds true in the wider population, this would underscore that the familiar 'motor homunculus' (*Penfield and Boldrey, 1937*) is overly simplistic. It is generally recognized that motor cortex does not rigidly follow a sequential point-to-point somatotopy, and indeed, Penfield and colleagues were aware of this and intended for their diagram to be a simplified summary of results showing partially overlapping motor fields that also varied substantially across individuals (*Catani, 2017*). However, the patchy mosaicism amongst nearby body parts in the current view of precentral gyrus organization still features a dorsal-to-ventral progression and separation of the major body regions (leg, arm, head) (*Farrell et al., 2007*; *Schieber, 2001*).

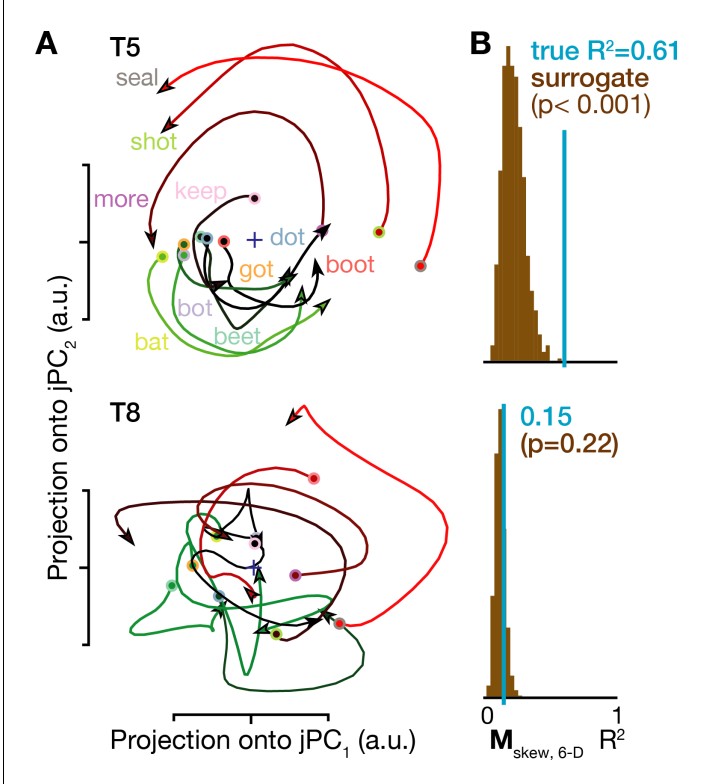

**Figure 5.** Rotatory neural population dynamics during speech. (**A**) The top six PCs of the trial-averaged firing rates from 150 ms before to 100 ms after acoustic onset in the T5-words and T8-words datasets were projected onto the first jPCA plane like in *Churchland et al. (2012)*. This plane captures 38% of T5's overall population firing rates variance, and rotatory dynamics fit the moment-by-moment neural state change with $R^2 = 0.81$ in this plane and 0.61 in the top 6 PCs. In T8, this plane captures 15% of neural variance, with a rotatory dynamics $R^2$ of only 0.32 in this plane and 0.15 in the top six PCs. (**B**) Statistical significance testing of rotatory neural dynamics during speaking. The blue vertical line shows the goodness of fit of explaining the evolution in the top six PC's neural state from moment to moment using a rotatory dynamical system. The brown histograms show the distributions of this same measurement for 1000 neural population control surrogate datasets generated using the tensor maximum entropy method of *Elsayed and Cunningham (2017)*. These shuffled datasets serve as null hypothesis distributions that have the same primary statistical structure (mean and covariance) as the original data across time, electrodes, and word conditions, but not the same higher order statistical structure (e.g. low-dimensional rotatory dynamics).

The presence of neurons responding to mouth and tongue movements in the dorsal 'arm and hand' area of motor cortex indicates that sensorimotor maps for different body parts are even more widespread and overlapping than previously thought. Given our previous finding that activity from these same arrays encodes intended arm and hand movements (*Pandarinath et al., 2017*), these observations are consistent with the hypothesis that the systems for speech and manual gestures are interlocked (*Gentilucci and Stefani, 2012*; *Rizzolatti and Arbib, 1998*; *Vainio et al., 2013*). However, emerging work from our group showing that neurons in this area also modulate during attempted movements of the neck and legs (*Willett et al., 2019*) suggests that much of the body is represented (to varying strengths) in dorsal motor cortex. Thus, the observed neural overlap between hand and speech articulators may be a consequence of distributed whole-body coding, rather than a privileged speech-manual linkage.

Our data suggest that the observed neural activity reflects movements of the speech articulators (the tongue, lips, jaw, and larynx): modulation was greater during speaking than after hearing the prompt; the same neural population modulated during non-speech orofacial movements; and in T5, the neural correlates of producing different phonemes grouped according to these phonemes' place of articulation. We also found that firing rates showed modest correlation with T5's unattended and

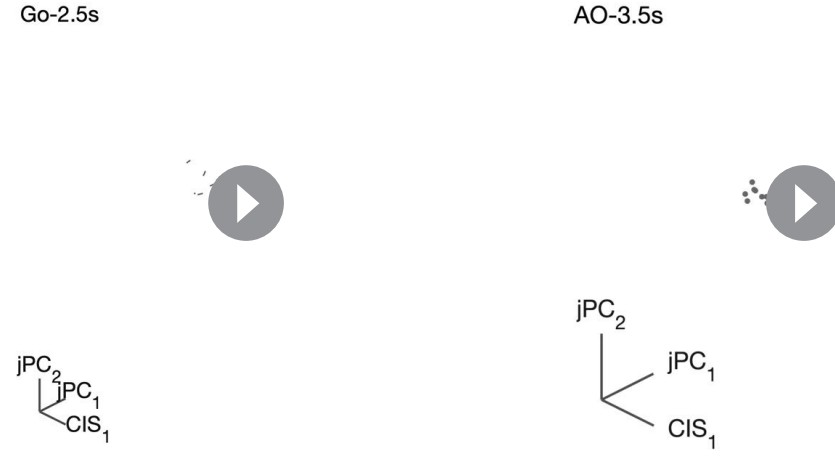

**Video 2.** The progression of neural population activity during the prompted words task is summarized with dimensionality reduction chosen to highlight the condition-invariant 'kick' after the go cue, followed by rotatory population dynamics. T5-words dataset neural state space trajectories are shown from 2.5 s before go cue to 2.0 s after go. Each trajectory corresponds to one word condition's trial-averaged firing rates, aligned to the go cue. The neural states are projected into a three-dimensional space consisting of the $CIS_1$ dimension (as in **Figure 4**) and the first two jPC dimensions (similar to **Figure 5**, except that for this visualization we enforced that the jPC plane be orthogonal to the $CIS_1$; see Materials and methods). The trajectories change color based on the task epoch: gray is before the audio prompt, blue is after the prompt, and then red-to-green is after the go cue, with conditions ordered as in **Figure 5**.
https://elifesciences.org/articles/46015#video2

**Video 3.** The same neural trajectories as **Video 2**, but aligned to acoustic on (AO), are shown from 3.5 s before AO to 1.0 s after AO.
https://elifesciences.org/articles/46015#video3

instructed breathing, which invites the question of how this activity relates to the precise control of breathing necessary for speaking and whether breath-related activity differs depending on behavioral context. A deeper understanding of how motor cortical spiking activity relates to complex speaking behavior will require future work connecting it to continuous articulatory (**Chartier et al., 2018**; **Conant et al., 2018**; **Mugler et al., 2018**) and respiratory kinematics and, ideally, the underlying muscle activations.

An important unanswered question, however, is to what extent these results were potentially influenced by cortical remapping due to tetraplegia. While we cannot rule this out, we believe that remapping of face representation to the hand knob area is unlikely. Despite these participants' many years of paralysis, the sites we recorded from still strongly modulate during attempted hand and arm movements (**Ajiboye et al., 2017**; **Brandman et al., 2018**; **Pandarinath et al., 2017**). We also verified in participant T5 that modulation during attempted arm movements was stronger than during speech production. Our ongoing work also indicates that this area modulates during attempts to move other body parts (e.g. the leg) which, like the arm, are also paralyzed (**Willett et al., 2019**). Taken together, these results are inconsistent with this area being 'taken over' by functions related to the participants' remaining capability to make orofacial movements. Furthermore, motor cortical remapping following arm amputation was recently shown to be smaller than previously thought (**Wesselink et al., 2019**), and in particular much smaller than what would be needed to move lip representations to hand cortex (**Makin et al., 2015**). On the sensory side, emerging evidence suggests that cortical reorganization following injury in adults is more limited than previously thought (**Makin and Bensmaia, 2017**), and a recent microstimulation study in the hand somatosensory cortex of a person with tetraplegia did not find functional reorganization (**Flesher et al., 2016**). While these threads of evidence argue against remapping, definitively resolving this ambiguity would require intracortical recording from this eloquent brain area in able-bodied people.

Assuming that these results are not due to injury-related remapping, we are left with the question of *why* this speech-related activity is found in dorsal 'arm and hand' motor cortex. Speech is spared following lesions in this area (*Chen et al., 2006*; *Tei, 1999*), indicating that it is not necessary for speech production. Nonetheless, it is possible that dorsal motor cortex plays some supporting role in speaking, perhaps contributing to more demanding speaking tasks, or that this activity reflects speech efference copy for coordinating orofacial and upper extremity movements. This would be in line with theoretical arguments that high dimensional representations resulting from mixed selectivity – in this case, both within major body regions (a given neuron being tuned for multiple arm movements or for multiple orofacial movements) and across major body regions (neurons being tuned for both arm and face movements) – enable more complex computations (*Fusi et al., 2016*) such as coordinating movements across the body. We anticipate that it will require substantial future work to understand why speech-related activity co-occurs in the same motor cortical area as arm and hand movement activity, but that this line of inquiry may reveal important principles of how sensorimotor control is distributed across the brain (*Musall et al., 2019*; *Stringer et al., 2019*).

Our second main finding is that, based on offline decoding results, intracortical recordings show promise as signal sources for BCIs to restore speech to people with some forms of anarthria. Decoding the neural correlates of attempted speech production (*Brumberg et al., 2011*) into audible sounds or text may be more desirable than approaches that decode covert internal speech (*Leuthardt et al., 2011*; *Martin et al., 2016*) or more abstract elements of language (*Chan et al., 2011*; *Yang et al., 2017*) because decoding attempted movements leverages existing neural machinery that separates internal monologue and speech preparation from intentional speaking. The present results compare favorably to previously published decoding accuracies using ECoG (*Mugler et al., 2014*; *Ramsey et al., 2018*) despite our dorsal recording locations likely being suboptimal for decoding speech. Multi-electrode arrays placed in ventral motor cortex would be expected to yield even better decoding accuracies. Furthermore, recent order-of-magnitude advances in the number of recording sites on intracortical probes (*Jun et al., 2017*) point to a path that stretches far forward in terms of scaling the number of distinct sources of information (neurons) for speech BCIs.

That said, these results are only a first step in establishing the feasibility of speech BCIs using intracortical electrode arrays. We decoded amongst a limited set of discrete syllables and words in participants who are able to speak; future studies will be needed to assess how well intracortical signals can be used to discriminate between a wider set of phonemes (*Brumberg et al., 2011*; *Mugler et al., 2014*), in the absence of overt speech (*Brumberg et al., 2011*; *Martin et al., 2016*), and to synthesize continuous speech (*Akbari et al., 2019*; *Anumanchipalli et al., 2019*; *Makin et al., 2019*). We also observed worse decoding performance in participant T8, highlighting the need for future studies in additional participants to sample the distribution of how much speech-related neural modulation can be expected, and what speech BCI performance these signals can support.

Our third main finding is that two motor cortical population dynamical motifs present during arm movements were also significant features of speech activity. We observed a large condition-invariant change at movement initiation in both participants, and rotatory dynamics during movement generation in the one of two participants whose arrays recorded substantially more modulation. We speculate that these neural state rotations are well-suited for generating descending muscle commands driving the out-and-back articulator movements that form the kinematic building blocks of speech (*Chartier et al., 2018*; *Mugler et al., 2018*). The presence of these dynamics during both reaching and speaking could indicate a conserved computational mechanism that is ubiquitously deployed across multiple behaviors to shift the circuit dynamics from withholding movement to generating the appropriate muscle commands from an oscillatory basis set. Testing and refining this hypothesis calls for examining whether these two dynamical motifs are present across an even wider range of behaviors and body parts. For instance, there is emerging evidence that rotatory dynamics may be absent in movements with a greater role of sensory feedback, such as hand grasping (*Suresh et al., 2019*).

This interpretation should also be tempered by the major unresolved question of whether these dynamics in dorsal motor cortex play a causal role in speaking and/or echo similar dynamics in other areas, such as ventral motor cortex, which are more directly involved in speech (*Bouchard et al., 2013*). An alternative interpretation is that if dorsal motor cortex merely receives an efference copy or 'coordination' signal about speech articulator movements, its dynamics may resemble those

during arm reaching because this is what the inherent properties of the local circuit are set up to generate – even if in the speech case, this activity is not helping construct muscle activities. Testing these hypotheses will require future research involving recording from the speech articulator muscles (analogous to recording from arm muscles in *Churchland et al., 2012*), causally stimulating the circuit (*Dichter et al., 2018*), and examining whether these neural ensemble dynamical motifs are present during speech production in ventral (speech) motor cortex.

## Materials and methods

### Participants

The two participants in this study were enrolled in the BrainGate2 Neural Interface System pilot clinical trial (ClinicalTrials.gov Identifier: NCT00912041). The overall purpose of the study is to obtain preliminary safety information and demonstrate proof of principle that an intracortical brain-computer interface can enable people with tetraplegia to communicate and control external devices. Permission for the study was granted by the U.S. Food and Drug Administration under an Investigational Device Exemption ( Caution: Investigational device. Limited by federal law to investigational use). The study was also approved by the Institutional Review Boards of Stanford University Medical Center ( protocol #20804), Brown University (# 0809992560), University Hospitals of Cleveland Medical Center (#04-12-17), Partners HealthCare and Massachusetts General Hospital (#2011P001036), and the Providence VA Medical Center (#2011–009). Both participants gave informed consent to the study and publications resulting from the research, including consent to publish photographs and audiovisual recordings of them.

Participant 'T5' (male, right-handed, 64 years old at the time of the study) was diagnosed with C4 AIS-C spinal cord injury 10 years prior to these research sessions. He retained the ability to weakly flex his left elbow and fingers and some slight and inconsistent residual movement of both the upper and lower extremities. T5 was able to speak normally and converse naturally without hearing assistance, but had some trouble hearing from his left ear.

Participant 'T8' (male, right-handed, 56 years old at the time of the study) was diagnosed with C4 AIS-A spinal cord injury 11 years prior to these sessions. He retained restricted and non-functional voluntary shoulder girdle motion on both sides, and non-functional voluntary finger extension on his left side. He had no sensation below the shoulder. T8 was able to speak normally and converse naturally with the assistance of hearing aids in both his ears.

### Prompted speaking tasks

Participants performed a syllables task consisting of discrete trials in which they spoke out loud one of 10 different phonemes or consonant-vowel syllables in response to an auditory prompt. These prompts were *i* (as in 'beet'); *ae* (as in 'bat'); *a* (as in 'bot'); *u* (as in 'boot'); *ba; da; ga; sh* (as in the start of 'shot'), and the unvoiced *k* and *p*. All pronunciations were American English. *Video 1* provides a continuous audio recording of one set of each type of syllables task trial.

Participants sat comfortably in a chair facing a microphone in a quiet room. They were instructed to refrain from attempting movements or speaking during trials except when prompted to speak by a custom experiment control software written in MATLAB (The Mathworks). During trials, they were also asked to fixate on the same object in front of them. Each trial began with two beeps to alert the participant that the trial was starting. Approximately 1 s after the start of the second beep, a pre-recorded syllable prompt was played via computer speakers. Two clicks played ~2 s after the start of the prompt served as the go cue that instructed the participant to speak back the prompted sound. The next trial started 2.8 s after the start of the second click. There was also an eleventh 'silent' condition which was identical to the spoken syllables trials, except that instead of playing a syllable prompt, the speakers played a nearly-silent audio file consisting of ambient background noise recorded in the same environment as the syllable prompts. The participants had been previously instructed not to say anything in response to this silent prompt.

The task was performed in blocks consisting of 10 trial sets. Each set contained 11 trials: one trial of each syllable, plus silence, presented in a randomized order. After the task was explained to each participant, he was given time to practice a few sets of the task until he indicated that he was ready to begin data collection. At the end of each set, we paused the task until the participant indicated

that he was ready to continue. These inter-set pauses typically lasted less than 10 s. Participants performed three consecutive blocks of the task during a research session, with longer pauses of several minutes between blocks during which we encouraged the participant to rest, adjust his posture for comfort, and take a drink of water.

Both the audio prompts played by the experiment control computer, and the participant's voice, were recorded by the microphone (Shure SM-58). This audio signal was recorded via the analog input port of the electrophysiology data acquisition system and digitized at 30 ksps together with the raw neural data (see Neural Recording section). Each trial's acoustic onset time (AO) was manually determined by visual and auditory inspection of the recorded audio data. During this review, we also excluded infrequent trials where the participant spoke at the wrong time or when the trial was interrupted (for example, if a caregiver entered the room). Isolated sounds can be difficult to discriminate, and our participants sometimes misheard a syllable prompt as a phonetically similar prompt. In particular, T5 misheard the majority of *da* prompts as *ga* (or occasionally as *ba*). Both participants made a few other substitutions between similar syllables. In this study, we were interested in the neural correlates of preparing and then generating speech, which should reflect the syllable that the participant perceived. We therefore labeled these misheard trials based on the spoken, rather than prompted, syllable for subsequent analyses. This left an insufficient number of T5 *da* trials for subsequent neural analyses; thus, there are 11 conditions shown in T8's *Figure 1* firing rate plots and *Figure 3* confusion matrices, but only 10 conditions for T5. The number of trials analyzed for each participant, after excluding trials and re-labeling misheard trials as described above, were: silent (30 trials for T5, 30 trials for T8); *i* (30, 28); *u* (30, 31); *ae* (28, 30); *a* (30, 30); *ba* (31, 29); *ga* (50, 34); *da* (0, 27); *k* (30, 27); *p* (30, 33); *sh* (30, 30). We refer to these datasets as 'T5-syllables' and 'T8-syllables'.

Participants also performed a words task which was identical to the syllables task except that they heard and repeated back one of 10 short words, rather than syllables, in response to the auditory prompt. Each participant performed three blocks of ten repetitions of each word during one research session. We refer to these datasets as 'T5-words' and 'T8-words'. Two consecutive trials were excluded from the T8-words dataset because of a large electrical noise artifact across almost all electrodes. The specific words, and the number of trials analyzed for each participant, were: 'beet' (30 T5 trials, 29 T8 trials); 'bat' (30, 29); 'bot' (30, 28); 'boot' (30, 30); 'dot' (30, 29); 'got' (29, 29); 'shot' (29, 28); 'keep' (30, 30); 'seal' (30, 30); 'more' (30, 30). As with the syllables task, there was also a silent condition (30 T5 trials, 30 T8 trials). During two additional research sessions (as part of a follow-up study), participant T5 performed the words task with only five of the 10 words. The conditions and trial counts in these two replication datasets, which we refer to as 'T5-5words-A' and 'T5-5words-B', were: 'seal' (33 trials in T5-5words-A, 34 trials in T5-5words-B); 'shot' (34, 34); 'more' (33, 34); 'bat' (34, 33); beet' (34, 34); and a silent condition (34, 34).

Silent condition trials were assigned a 'faux AO' so that neural data from comparable epochs of silent and spoken trials could be visualized and analyzed (for example, for generating trial-averaged, AO-aligned firing rates in *Figure 1* or for decoding silent trials' neural activity in *Figure 3*). Specifically, each silent trial's AO was set to equal the mean AO (relative to the go cue) for all the spoken syllables or words during the same block.

## Orofacial movement task

Participants also performed an orofacial movement task with a similar trial structure as the syllables and words tasks. Seven different movement conditions were instructed with auditory prompts: 'mouth open', 'lips forward', 'lips back', 'tongue right', 'tongue down', 'tongue up', and 'tongue left'. An additional 'stay still' condition was analogous to the silent condition of the syllables and words tasks. Prior to the first block of the orofacial task, a researcher explained the prompts to the participant, demonstrated the movements, and ran the participant through a few practice sets. Due to clinical trial protocols, we did not collect kinematic tracking data such as electromagnetic midsagittal articulography (*Chartier et al., 2018*) or ultrasound recordings (*Conant et al., 2018*). A video recording of the participants' faces (without markers) did allow the researchers to confirm that the participants were making the instructed movement with acceptable timing precision. Given this limitation, we limited our use of these data to broadly testing for neural responses during orofacial movements, rather than quantifying precise moment-by-moment relationships between neural activity and kinematics.

Similar to the syllables and words task, an orofacial movement trial began with two ready beeps, after which the computer speaker played a movement prompt (e.g. 'lips forward'). This was followed by the pair of go clicks; the participants were previously informed that they should begin moving after the second click. Approximately 1.9 s after the go cue click, the experiment control system played the verbal command 'return', which instructed the participant to return to a neutral orofacial posture (e.g. close the mouth after 'mouth open', move the tongue left after 'tongue right'). The trial ended ~1.9 s after the start of 'return'. The purpose of using a return cue was so that there was a known epoch after the movement go cue during which we knew that the participant was not yet returning. The return cue also provided the participant with dedicated time to return to a neutral orofacial position, so that all trials would start from roughly the same posture. For T8, the 'return' instruction was immediately followed by a go click. However, we observed that T8 started the return movement upon hearing 'return' rather than waiting for the go click. We therefore removed the return go click prior to T5's research sessions, and instead instructed T5 to start the return movement when he heard 'return'. In the present study, we did not examine the return portion of the orofacial movement task.

Each participant's orofacial movements and syllables datasets were collected on the same day during the same research session; three blocks of the orofacial movement task immediately followed three blocks of the syllables task. We will refer to these orofacial movements task datasets as 'T5-movements' and 'T8-movements'. No trials were excluded from these datasets; thus, there were 30 trials of each condition for each participant.

## Many words task

During an additional research session, participant T5 performed a many words task in which he spoke 420 unique words (from *Angrick et al., 2019*) designed to broadly sample American English phonemes. These words were visually prompted, with one word appearing per trial. Each trial started with an instruction period in which a red square appeared in the center of a computer screen facing the participant. White text above the square instructed what word the participant should say once given a go cue (e.g. 'Prepare: 'Dog''). This instruction period lasted 1.2 to 1.8 s (mean 1.4 s, exponential distribution) after which the square turned green, the text changed to 'Go', and an audible beep was played. This served as the go cue for T5 to speak out loud the instructed word. A second beep occurred 1.5 s later, which marked the end of the trial. The next trial began 1 s later. The 420 words were divided into four sets, with each set spoken during a continuous block of trials with short breaks between blocks. Each word set was repeated three times during this research session, with a given set's words appearing in a different random order during each block. We call this the 'T5-phonemes' dataset.

## Breath measurement

T5's breath-related abdomen movements were measured with a piezo respiratory belt transducer (model MLT1132, ADIntruments). The stretch sensor was wrapped around his abdomen at the point where it maximally expanded during breathing. Analog voltage signals from the belt were input to the neural signal processor via one of its analog input channels. These data were digitized at 30 ksps along with the neural data. Our goal was to test whether there is breath-correlated neural activity during 'unattended' breathing (i.e. natural 'background' breathing, when the participant was not consciously attending to his breath) and during consciously attended 'instructed' breathing. Both of the unattended and instructed conditions were collected during the same research session, and we refer to this as the 'T5-breathing' dataset.

For the unattended breathing condition, we recorded neural and breath proxy measurements while T5 performed a BCI computer cursor task as part of a different study, and during an interval where he was resting quietly after completing the BCI task. For the instructed breathing task, we recorded neural and breath proxy measurements while T5 performed a cued breathing task that followed a similar structure as the many words task described in the previous section. On each trial, the on-screen instruction text was either 'Prepare: Breathe in' or 'Prepare: Breathe out'. The order of these two trial types was randomized within consecutive two-trial sets, such that breaths in and breaths out were counterbalanced and no more than two out breaths or two in breaths could be prompted in a row. After a random delay of 1.2 to 1.6 s (mean 1.4 s, exponential distribution), the

go cue instructed the participant to breathe in or out according to the instruction. After 1.5 s, an audible beep and the on-screen text changing to 'Return' instructed the participant to return to a neural lung inflation position. 'Return' stayed on screen for 1.5 s, after which the inter-trial interval was 1 s. A block consisted of 12 trials, after which the participant was given a chance to take a break, relax, and breathe naturally before the next block. The participant reported that this task was comfortable and that he was able to match his breaths to the instructions without difficulty.

## Movement comparisons task

The purpose of this task, which was performed on a separate day from the other datasets, was to compare the neural modulation when making orofacial movements and speaking, versus when attempting to make arm and hand movements. The task had a similar visually instructed structure to the instructed breathing task. During the instructed delay period, text displayed the upcoming movement, for example, 'Prepare: Say Ba', or 'Prepare: Open Hand'. There was also a 'Prepare: Do Nothing' instruction, which otherwise had the same trial structure as the instructed movements. After a random delay period of between 1400 and 1800 ms, the go cue appeared. During this epoch, T5 attempted to make the instructed movement as best as he could. This resulted in complete movements for all the orofacial and speaking movements and 'shoulder shrug', partial movements for some of the arm movements (e.g. 'flex elbow in'), and no overt movement for the other arm movements (e.g. 'close hand', 'thumb up'). We analyzed neural data from 200 ms to 600 ms after the go cue. We note that insofar as there was somatosensory and proprioceptive feedback only during the actualized movements, this would be expected to increase the observed neural modulation to orofacial movements and speaking, and decrease the modulation to attempted arm and hand movements. The go cue stayed on for 1500 ms. This was followed by a return period in which the text changed to 'Return'; during this epoch, the participant was instructed to return his body to a neutral posture. Thirty-two trials were collected for each movement type. We refer to this as the 'T5-comparisons' dataset.

## Neural recording

Both participants had two 96-electrode Utah arrays (1.5 mm electrode length, Blackrock Microsystems) neurosurgically placed in dorsal 'hand knob' area of the left (motor dominant) hemisphere's motor cortex. Surgical targeting was stereotactically guided based on prior functional and structural imaging (*Yousry et al., 1997*), and subsequently confirmed by review of intra-operative photographs. T5 and T8 had arrays placed 14 and 34 months, respectively, prior to the present study's prompted words, syllables, and orofacial movements tasks. The T5-breathing and T5-comparisons datasets were recorded 26 months after array placement, the T5-5words-A and T5-5words-B datasets were recorded 28 months after array placement, and the T5-phonemes dataset was recorded 29 months after array placement. Arrays were placed in areas anticipated to have arm movement-related activity because two goals of the clinical trial are 1) testing the feasibility of intracortical BCI-based communication using point-and-click keyboards and 2) restoration of reach and grasp function via control of a robotic arm or functional electrical stimulation. We note that these implant sites are distinct from the closest known speech area, which is the dorsal laryngeal motor cortex (*Bouchard et al., 2013*; *Dichter et al., 2018*). In this study, we looked for neural correlates of speaking in dorsal motor cortex. To help contextualize the results, here we summarize the other behaviors associated with modulation of the neural activity recorded by these same arrays. Our previous studies have reported that T5 and T8 controlled BCI computer cursors by attempting movements of their arm and hand (*Brandman et al., 2018*; *Pandarinath et al., 2017*). T8 was also able to use intended arm movements to command movements of his own paralyzed arm via functional electrical stimulation (*Ajiboye et al., 2017*). We also recorded movement task outcome error signals from T5's arrays; these signals indicated whether the participant succeeded or failed at acquiring a target using a BCI-controlled cursor (*Even-Chen et al., 2018*).

Neural signals were recorded from the arrays using the NeuroPort system (Blackrock Microsystems). Voltage was measured between each of the 96 electrodes' uninsulated tips and that array's reference wire. Wire bundles ran from each array to cranially-implanted connector pedestals. During research sessions, a 'patient cable' with a unity gain pre-amplifier was connected to each array's corresponding pedestal and carried signals to an isolated unity gain front-end amplifier. These signals

were analog filtered from 0.3 Hz to 7.5 kHz, digitized at 30 kHz (250 nV resolution), and sent to the neural signal processor via fiber-optic link. As mentioned earlier, amplified analog voltage data from the microphone were input to the neural signal processor and were digitized time-locked with the neural signals. All these digitized data were sent over a local network to a connected PC where they were recorded to disk for subsequent analysis.

The naming scheme for neurons or electrodes in figures is <participant>_<array #>.<electrode #>. For example, 'neuron T5_2.4' in *Figure 1* refers to a participant T5 neuron identified on the second array (which is the more medial of each participant's two arrays) on electrode #4 (according to the manufacturer's electrode numbering scheme).

For both participants, we did not observe major differences between the two arrays, and we confirmed that the neural population analyses results (ensemble modulation to speech/movements/breathing, phoneme neural correlate similarities, speech decoding, condition-invariant and rotatory population dynamics) were similar when data from each array were analyzed separately. We therefore pool together data from both arrays in all the presented results.

## Neural signal processing

Neuronal action potentials (spikes) were detected as follows. We first applied a common average re-referencing to each electrode within an array by subtracting, at each time sample, the mean voltage across all electrodes on that array. These voltage signals were then filtered with a 250 Hz asymmetric FIR high-pass filter designed to extract spike activity from this type of array (*Masse et al., 2014*). To measure single unit activity (SUA), time-varying voltages were manually 'spike sorted' by an experienced neurophysiologist using Plexon Offline Spike Sorter v3. This process identified action potentials belonging to putative individual neurons amongst the high amplitude voltage deviation events. Occasionally, the same action potential can be recorded on multiple electrodes (this could happen if a neuron is very large, if an axon passes multiple electrodes, or if there is some electrical cross-talk in the recording hardware). To prevent creating duplicate single neuron units, we excluded 'cross-talk units' if their spike time series (using 1 ms binning) had greater than 0.5 correlation with another unit's. When this happened, we kept the unit with the better spike sorting isolation. Unless otherwise stated, time-varying firing rate plots, also known as peristimulus time histograms (such as in *Figure 1D*) were constructed by smoothing spike trains with a 25 ms s.d. Gaussian kernel and averaging continuous-valued firing rates across trials of the same behavioral condition.

Spike sorting allows us to make statements about the properties of individual motor cortical neurons (for example, how many syllables they modulate to, as in *Figure 1—figure supplement 4B*). However, a limitation of spike sorting is that action potential event 'clusters' with insufficient isolation from other clusters are discarded. For chronic multielectrode array recordings, this can mean that activity recorded from the majority of electrodes is not analyzed, despite these neural signals having a strong relationship with the behavior of interest. This problem is particularly acute in human neuroscience, where replacing arrays, or using newer methods that provide a higher SUA yield (for example high-density probes or optical imaging), is not currently possible. Relaxing the constraint that action potential events must be unambiguously from the same neuron and instead analyzing voltage threshold crossings (TCs) is an effective way to substantially increase the information yield of chronic electrode arrays. In this study, we examined TCs in a number of analyses. Decoding TCs or other non-SUA signals has become standard practice in the intracortical BCI field (e.g. *Ajiboye et al., 2017*; *Brandman et al., 2018*; *Collinger et al., 2013*; *Even-Chen et al., 2018*; *Pandarinath et al., 2017*). This method also provides information about the dynamics of the neural state (i.e. it can be used to make scientific statements about ensemble activity under many conditions) despite combining spikes that may arise from one or more neurons; we provide empirical and theoretical justifications in *Trautmann et al. (2019)*. In the present study, when we refer to an 'electrode's' firing rate, we mean TCs recorded from that electrode. When we refer to a neuron's firing rate, we mean sorted single unit activity. *Figure 1—figure supplement 2* shows example TCs firing rates, including from the same electrodes that the example neurons in *Figure 1* were sorted from.

A threshold of $-4.5 \times$ root mean square (RMS) voltage was used for all analyses and visualizations except for the t-SNE visualization and decoding analyses shown in *Figure 3*. This threshold choice is somewhat arbitrary but is conservative; it accepts large voltage deviations indicative of action potentials from one or a few neurons near the electrode tip. For the *Figure 3* analyses, we used a more relaxed threshold of $-3.5 \times$ RMS because we found that this led to slightly better classification

performance in a separate pilot dataset (consisting of T5 speaking five words and syllables, collected a month prior to the datasets reported here) which we used for choosing hyperparameters. The better performance of a less restrictive voltage threshold is consistent with collecting more information by accepting spikes from a potentially larger pool of neurons (*Oby et al., 2016*). This trade-off was acceptable because for these engineering-minded decoding analyses, we were less concerned about the possibility of missing tuning selectivity or fast firing rate details due to combining spikes from more neurons.

Electrodes with TCs firing rates of less than 1 Hz (at a $-4.5 \times$ RMS threshold) were considered non-functioning and were excluded from analyses unless there was well-isolated SUA on the electrode. This electrode exclusion applied to both spikes and the local field potential signal described below. Electrodes having TCs time series with greater than 0.5 correlation with another electrode's were marked for cross-talk de-duplication. To determine which electrode to keep, we chose the one that had the fewest spikes co-occurring (1 ms bins) with the other electrode(s)' (i.e. we kept the electrode with putatively more unique information).

For the neural decoding analyses (*Figure 3*), we also extracted a high-frequency local field potential (HLFP) feature from each electrode by taking the power of the voltage after filtering from 125 to 5000 Hz (third-order bandpass Butterworth causal filtering forward in time). HLFP is believed to contain substantial power from action potentials (*Waldert et al., 2013*); we view this feature as capturing spiking 'hash', that is multiunit activity local to the electrode with contributions from smaller-amplitude and more distant action potentials than TCs. Our previous study found that this signal is highly informative about hand movement intentions and is useful for real-time BCI applications (*Pandarinath et al., 2017*). This feature has some similarities to the 'high gamma' activity examined by ECoG studies; the definition of high gamma varies in exact frequency from study to study, but generally has a lower cutoff between 65 and 85 Hz and an upper cutoff between 125 and 250 Hz (*Bouchard et al., 2013*; *Chartier et al., 2018*; *Cheung et al., 2016*; *Dichter et al., 2018*; *Martin et al., 2014*; *Mugler et al., 2014*; *Ramsey et al., 2018*). However, the intracortical HLFP in this study should not be viewed as being the exact same as ECoG high gamma activity due to differences in electrode location, electrode geometry, and HLFP's higher frequency range.

## Task-related neural modulation

To quantify which electrodes' spiking activity changed during speaking (*Figure 1B* insets, *Figure 1—figure supplement 4*), we calculated each electrode's mean firing rate from 0.5 s before to 0.5 s after AO, yielding one datum per electrode, per trial. For each syllable, a rank-sum test was then used to determine whether there was a significant change in the distribution of single-trial firing rates when speaking the syllable compared to the silent condition (p<0.05, Bonferroni corrected for the number of syllables). To identify which electrodes responded to orofacial movements (*Figure 2*, *Figure 2—figure supplement 1*) we performed a similar analysis, except that the analysis epoch was from 0.5 s before to 0.5 s after the go cue. This epoch captures strong modulation, as can be seen by the example firing rate plots in *Figure 2*. We note that firing rate changes preceding the go cue indicate either substantial movement preparation activity, or that the participants were 'jumping the gun' and started moving in anticipation of the go cue; either way, this response indicates modulation related to making orofacial movements. In lieu of a silent condition, the movement conditions' firing rate distributions were compared to that of the 'stay still' condition. The same methods were used to quantify which single neurons' activities changed during speaking or orofacial movements; for this, we analyzed SUA rather than electrodes' $-4.5 \times$ RMS TCs.

## Neural population modulation

To measure the differences in neural modulation across the recorded population following the audio prompt and following the go cue ('population modulation' in *Figure 1E*, *Figure 1—figure supplement 3B*), at each time point (aligned to either the audio prompt or the go cue) we quantified the differences between the firing rate vector for a given spoken condition $\mathbf{y}_{speak}$ (for example, the vector of firing rates across the *ga* syllable trials, where each element of the vector is the firing rate for one electrode) and $\mathbf{y}_{silent}$, the firing rate vector for the silent condition. Importantly, however, we did not simply use $\|\mathbf{y}_{speak} - \mathbf{y}_{silent}\|$, the Euclidean norm of the vector difference between these two conditions' trial-averaged firing rates. The problem with that approach is that a vector norm always

yields a non-negative value, meaning that if it is used to measure neural activity differences, the metric will be upwardly biased: it will return a positive value instead of 0 even when population firing rates for the two conditions are essentially the same. This is because estimates of firing rates for two sets of trials, even if they are drawn from the same underlying distribution (i.e. from the same behavioral context) will inevitably differ, even just slightly, resulting in a positive vector difference norm. This problem becomes worse when dealing with lower trial counts and low firing rates, and makes it difficult to distinguish weak population modulation from noise.

To avoid this issue and better estimate neural population activity differences, we used a cross-validated variant of the vector difference norm; we will refer to this metric as the 'neural distance'. For $N_1$ trials from condition 1 (for example, saying *ga*) and $N_2$ trials from condition 2 (for example, silent trials), we calculate a less biased estimate of the squared vector norm of the difference in the two conditions' mean firing rates using:

$$D = \frac{1}{N_1}\frac{1}{N_2}\sum_{i=1}^{N_1}\sum_{j=1}^{N_2}\left[\boldsymbol{y}_1^i - \boldsymbol{y}_2^j\right]^T \cdot \left[\boldsymbol{y}_1^{\{1:N_1\}/i} - \boldsymbol{y}_2^{\{1:N_2\}/j}\right] \tag{1}$$

which leaves one trial out from each condition when calculating the differences in means. Critically, the dot product is taken between firing rates computed from fully non-overlapping sets of trials, and can be negative. To convert this to a signed distance more analogous to a Euclidean vector norm, we define the final neural distance metric as $d = sign(D)\sqrt{|D|}$.

This cross-validated neural distance has units of Hz; much like with a standard Euclidean vector norm, having more electrodes, and these electrodes having larger firing rate differences between the two conditions, will both result in larger overall distances. Unlike a Euclidean vector norm, our population neural distance metric can produce negative values. This is required for the metric to be unbiased and should be interpreted as evidence that the true distance between the two distributions' population firing rates is near zero. A benefit of allowing negative values is that time-averaging across an epoch of essentially no underlying firing rate differences will give a mean distance close to zero. The derivation of this metric is described in detail in *Willett et al. (2019)*, and a software implementation is available at https://github.com/fwillett/cvVectorStats.

For statistical testing, we compared the time-average of this neural distance across two comparison epochs: a prompt epoch (0 to 1 s after the audio prompt) and a speaking epoch (0 to 1.75 s after the go cue for T5, 0.5 to 1.75 s after go for T8). We chose a later speaking epoch start for T8 to better match this participant's delayed speech-related modulation, which could reflect less anticipatory preparation prior to the (predictable) go cue time, and/or the reduced speech-related modulation recorded on T8's arrays. This resulted in one datum for each epoch per speech condition, for example 10 pairs of (prompt, speech) value pairs corresponding to each syllable. We compared the resulting prompt and speech epoch distributions with a Wilcoxon signed-rank test. The same procedure was used to compare the prompt epoch neural population modulation to a 'baseline' epoch consisting of the 1 s leading up to the audio prompt.

When we report the ratio between population modulation during the go epoch and during the prompt epoch, this ratio was computed after taking the mean modulation across all syllables/words for each epoch.

## Comparing different phonemes' neural correlates

To generate *Figure 1—figure supplement 5*, we first manually segmented each word spoken in the T5-phonemes dataset into its constituent phonemes using the Praat software package (*Boersma and Weenink, 2019*). This resulted in 3892 total phonemes. The number of occurrences across the 41 unique phonemes ranged between 14 (/ɔ/) and 239 (/t/), with a median of 80 occurrences. For each unique phoneme, we isolated a 150 ms window of TCs centered around the onset of each instance of that phoneme. This produced an (# instances) $\times$ electrodes firing rate matrix for each phoneme. We used these data matrices to calculate the neural population activity difference between all pairs of phonemes using the cross-validated neural distance metric described in the 'Neural population modulation' section. This resulted in the matrix of phoneme pair neural distances in *Figure 1—figure supplement 5A*. Within-phoneme neural distances (the diagonal elements of the distance matrix) were calculated by comparing half of the instances of a given phoneme with the

other half; the distances shown are the mean distances across 20 such random splits of each phoneme.

To relate these neural distances to known differences in the speech articulator movements required to produce the phonemes, we grouped phonemes by their place of articulation as in *Moses et al. (2019)*. We then compared within-group neural distances to between-groups neural distances (*Figure 1—figure supplement 5B*). Every pair of phonemes in the *Figure 1—figure supplement 5A* neural distance matrix contributes one datum to either the red distribution in that figure's panel B (if the two phonemes are in the same articulatory grouping) or to the black distribution (if the two phonemes are in different groups). The exception to this is that the three phonemes that are sole members of their own lonely groups were not included in this analysis. The summary statistic of this comparison was the difference between the mean of within-group neural distances and the mean of between-groups neural distances. This statistic was compared against a null distribution built by taking the same summary metric after shuffling neural distance matrix rows and columns, repeated 10,000 times. This null distribution assumes that the phonemes are grouped arbitrarily (but with the same number and sizes of groups), and not according to place of articulation. Comparing the true within-group versus between-groups difference to this null distribution (*Figure 1—figure supplement 5C*) provides a p-value for rejecting the null hypothesis that phoneme neural distances are no more correlated with articulatory grouping than expected by chance.

The dendrogram shown in *Figure 1—figure supplement 5D* was generated by applying the widely used 'unweighted pair group method with arithmetic mean' (UPGMA) hierarchical clustering algorithm (*Sokal and Michener, 1958*) to the phoneme neural distance matrix.

## Breath-related neural modulation

To generate breath-triggered firing rates (*Figure 2—figure supplement 2*), we first identified breath peak times from the breath belt stretch transducer measurements. The belt signals were pre-processed by removing rare outlier values (>50 µV difference between consecutive samples) and then low-pass filtering (3 Hz pass-band) the signal both forwards and backwards in time to avoid introducing a phase shift. An example of this filtered signal is shown in *Figure 2—figure supplement 2A*. Breath peaks were then found using the MATLAB *findpeaks* function, with key parameters of MinPeakDistance = 1 s, and MinPeakProminence = 0.3·B, where B is the median of all peak prominences found by first running *findpeaks* using MinPeakDistance = 5 s (in other words, we required a peak to be at least 30% of the prominence of the 'big' peaks in the data).

Breath peak-aligned firing rates were calculated by treating each identified breath peak as one trial, and trial averaging across neural snippets aligned to each breath peak time. Each TCs' or SUA's breath-related modulation depth was defined as the maximum – minimum firing rate observed in the interval from 2 s before the breath peak to 1.5 s after the breath peak. To calculate whether a given modulation depth was statistically significant, we used a shuffle control in which we compared the true data's modulation depth to the distribution of modulation depths observed over 1001 random shuffles in which faux peak breath times were uniformly drawn from the data duration. For comparing breath-related and speaking-related modulation depths (*Figure 2—figure supplement 2F*), we defined a given electrode's speech modulation depth in the T5-syllables dataset as its maximum – minimum firing rate from 2.5 s before acoustic onset to 1 s after acoustic onset.

## Arm and hand versus orofacial and speaking movements comparisons

The neural ensemble modulation comparisons presented in *Figure 2—figure supplement 3* were calculated as follows: mean TCs firing rates for each T5-comparisons dataset instructed movement condition were calculated for each electrode from 200 to 600 ms after the go cue. The resulting firing rate vectors were compared to firing rate vectors similarly constructed from the 'do nothing' condition. Modulation was calculated by taking the unbiased neural distance between these firing rate vectors as described above in the 'Neural population modulation' section.

## Single-trial low-dimensional neural state projections

To visualize single-trial high dimensional neural data (*Figure 3A*), we used t-distributed stochastic neighbor embedding (tSNE), a dimensionally reduction technique which seeks to represent high-dimensional vectors (such as our time-varying, multielectrode neural data) in a low-dimensional

space (such as a 2D plot that can be easily visualized). The tSNE algorithm finds a nonlinear mapping such that similar high-dimensional feature vectors end up close together in the low-dimensional view, while dissimilar vectors end up far apart (*Van Der Maaten and Hinton, 2008*). A neural feature vector was constructed for each trial as follows: for each functioning electrode, spike rates and HLFP power were calculated in ten 100 ms bins that spanned from 0.5 s before to 0.5 s after AO. These features were concatenated into a vector; for example, for the T5-syllables dataset, a single trial's neural data were represented as a 104 electrodes × 2 features per electrode ×10 time bins = 2080 dimensional vector. All trials' feature vectors were then projected into a 2D space using the *tsne* function in MATLAB R2017b's Statistics and Machine Learning Toolbox with NumDimensions = 2; Perplexity = 15 (this is the number of local neighbors examined for each datum); Algorithm = exact (suitable for our relatively small dataset); and Standardize = true (this z-scores the input data, which was desirable due to the variability between different electrodes and the vastly different scales between spike rates and HLFP power). All other algorithm parameters were set to their defaults. *Figure 3A* does not have axis labels because t-SNE does not return meaningful axes or units; only the relative distances between points have meaning.

## Speech decoding

We evaluated how well the identity of the syllable or word being spoken could be decoded from neural data by classifying single trial neural data. Neural feature vectors were constructed for each trial as described above. These vectors were then associated with a class label, which was the sound being spoken (i.e. word, syllable, or silence). We trained support vector machines (SVMs), a standard classification tool, to predict the class label from a 'new' neural feature vector which the classifier had not been trained on. Prediction accuracies were cross-validated using a leave-one-trial-out paradigm in which the classifier was trained on all trials except the trial being classified, and this was repeated for all trials in a dataset. Multiclass classification was achieved using the error-correcting output code (ECOC) technique, which trains multiple binary SVMs between all pairs of labels, that is a one-versus-one coding design. When classifying new input data, the ECOC technique picks the class that minimizes the sum of losses over the set of binary SVM classifiers. Specifically, we used MATLAB R2017b's implementation: a multiclass model object was fit (*fitcecoc*) using the SVM template (*templateSVM*). Key parameters were to use a linear kernel; OutlierFraction = 0.05 (expecting 5% of data points to be outliers); and Standardize = true (which z-scores the neural features based on the training data). All other parameters were set to their default values. We note that we did not heavily optimize our classification method; rather, our goal here was to use a standard tool to gauge the classification performance that these intracortical neural signals support. More sophisticated machine learning techniques (e.g. *Angrick et al., 2019*; *Livezey et al., 2019*) are likely to provide additional improvements.

To measure chance prediction performance, we used a shuffle test in which we randomly permuted the class labels associated with all trials' neural data. The same classifier training and leave-one-out prediction process was then repeated on these shuffled data 101 times.

## Neural population dynamics

An underlying motivation for the neural population dynamics analyses described in the next several sections is the idea that the activity of many thousands or millions of neurons in a circuit (of which we can only measure on the order of 100 neurons in humans with current technology) can be summarized by the time-varying activity of a handful of latent 'components'. In this framing, individual neurons' firing rates reflect various mixtures of these underlying components; in all the analyses we used, this mapping from components to firing rates is assumed to be linear. These components are not meant as discrete physical 'things' in the brain, but rather are mathematical abstractions which capture meaningful patterns in the activities of networks of neurons. They are useful insofar as they can help generate hypotheses about the computations neural populations are performing by describing their prominent activity patterns. To this end, not only can latent components succinctly describe the 'neural state' (i.e. the firing rate of all neurons at a given moment in time), but furthermore, the time evolution of these components is often more conducive to interpretation and understanding than more complex descriptions of all the individual neurons' firing rates.

Here, we built on previous studies showing that these components' changes over time can be effectively modeled as a lawful time-varying oscillatory dynamical system (*Churchland et al., 2012*; *Pandarinath et al., 2015*), and that they reveal a simple population-level pattern in which there is a stereotyped response at the initiation of many different movements (*Kaufman et al., 2016*). This 'dynamical system' framework is extensively reviewed in *Shenoy et al. (2013)* as well in the two key studies that inspire the neural population dynamics analyses of the present study (*Churchland et al., 2012*; *Kaufman et al., 2016*). We looked for the aforementioned dynamical motifs using two different dimensionality reduction techniques that were specifically designed to reveal the presence (or absence) of these population dynamics features.

For these analyses, we primarily examined the prompted word speaking task datasets because this was a more naturalistic behavior than the prompted syllables speaking task. Participants reported that it was more difficult to discriminate syllables than words, and that speaking stand-alone syllables felt somewhat awkward, whereas saying words was easy. Consequently, a practical benefit of the words task over the syllables task is that behavior was more stereotyped across trials, which facilitates precise trial-averaging, and there were very few mis-heard or mis-spoken words. Results for the same analyses applied to the syllables task data are shown in *Figure 4—figure supplement 1*.

Both of these neural population state analyses were performed on TCs, which contained more information about the neural population state than the more limited number of recorded SUA. All electrodes with TCs firing rates greater than 1 Hz were included. The Churchland-Cunningham and Kaufman studies analyzed a combination of both SUA from single-electrode recordings and TCs from multielectrode recordings, depending on the dataset, while *Pandarinath et al. (2015)* also analyzed just TCs. To avoid cumbersome switching of terms when describing our methods and comparing them to those of these previous studies, we will use the generic term 'unit' to refer to a single channel of neural information, whether it be SUA or TCs.

## Condition-invariant signal

The first population dynamics motif we tested for was a specific form of population-level structure at the initiation of movement: a large condition-invariant signal, previously described in *Kaufman et al. (2016)*. We closely followed Kaufman and colleagues' analysis methods, adapting them as necessary for these human speaking datasets. As in *Kaufman et al. (2016)*, spike trains were trial-averaged within a behavioral condition (in our case, speaking one of the 10 different words), smoothed with a 28 ms s.d. Gaussian, and 'soft normalized' with a 5 Hz offset. Normalization means that each unit's firing rate was normalized by its range across all times and conditions. This prevents units with very high firing rates from dominating the estimate of neural population state (*Pandarinath et al., 2018*). The 'soft' refers to adding an offset (5 Hz in these analyses) to the denominator to reduce the influence of units with very small modulation. Trial-averaged firing rates were calculated from a speech initiation epoch of 200 ms before go cue to 400 ms after the go cue for T5, and 100 ms to 700 ms after the go cue for T8. T8's epoch was shifted later relative to T5's to account for T8's later neural population activity divergence from the silent condition (*Figure 1—figure supplement 3B*). This yields a $N \times C \times T$ data tensor, where N is the number of units, C is the number of word conditions (10), and T is the number of time samples (600, using 1 ms sliding bins).

We used demixed principal components analysis (dPCA), a dimensionality-reduction technique developed by *Kobak et al. (2016)*, to look for condition-invariant activity patterns in these high-dimensional neural recordings. This dimensionality reduction method is conceptually similar to PCA, in that it finds a specified number of dPC 'components' that can be thought of as 'building blocks' from which the responses of individual units can be composed. As with PCA, dPCA attempts to compress the data by identifying dimensions that capture a large fraction of the variance. This takes advantage of the fact that unless the responses of neurons are all independent from one another (which in practice is not the case), then most of the variance of the full population response can be accurately reconstructed as a weighted sum of a smaller number of dPC components. Where dPCA differs from PCA is that it can explicitly attempt to find components that marginalize variance attributable to different parameters of the experiment (such as time or task variables). This is possible because dPCA is a supervised method that trades off finding dimensions that maximize variance in favor of finding dimensions that partition the variance based on labeled properties of the data.

In our case, this 'demixing' was attempted between: 1) condition and condition + time interactions, which together form the condition-dependent (CD) components of the neural population activity; and 2) time only, which forms condition-invariant (CI) components. In other words, dPCA sought a set of components of the population activity for which the time-varying neural responses during producing different words look the same, and also for another set of components which vary across speaking conditions (i.e. are 'tuned' for what word is being spoken). Importantly, such variance marginalization (i.e. demixing the parameters) may not be achievable; it depends on the structure of the data itself. Each component that dPCA returns is associated both with how much overall neural variance it captures (the lengths of the bars in *Figure 4A*), and how much of this variance is CI or CD (red and blue fraction of each bar, respectively). Thus, the success of this demixing can be examined based on how purely CI or CD each component is. This in turn reveals whether there exists a large and almost completely condition-invariant component of the population neural activity.

Kaufman and colleagues used an earlier version of the dPCA method and code package, called 'dPCA-2011' (*Brendel et al., 2011*). We used the MATLAB implementation of 'dPCA-2015' (*Kobak et al., 2016*), downloaded from https://github.com/machenslab/dPCA. This is an updated, improved, and widely adopted version of the technique which was not yet available at the time when the *Kaufman et al. (2016)* analyses were performed. We specified that dPCA should return eight total components, which was less than then 10 to 12 used in *Kaufman et al. (2016)*. This reflects the reduced complexity of our datasets, in the sense that they had fewer conditions (10 versus 27–108) and fewer units (96–106 versus 116–213). We also repeated the analyses using 2 to 12 dPCs and observed very similar results. Default *dpca* function parameters were used, with parameters numRep = 10 (repetitions for regularization cross-validation) and simultaneous = true (indicating that the single-trial neural data were simultaneously recorded across electrodes) for the *dpca_optimizeLambda* and *dpca_getNoiseCovariance* functions.

Unlike the dPCA-2011 used by *Kaufman et al. (2016)*, dPCA-2015 does not enforce that the neural dimensions found for capturing variance attributable to different parameters (here, the CI and CD components) be orthogonal. For example, while the first three (largely CI) components for T5 in *Figure 4A* are orthogonal by construction (as are the five largely CD components), these CI and CD components need not be orthogonal. We quantified the angles between the demixed principal axes (the dPCA encoder dimensions), and the (related but distinct degree of correlation between the resulting dPCA components, using the methods described in *Kobak et al. (2016)* and implemented in the dPCA code pack. Unlike *Kobak et al. (2016)*, we used a p-value threshold of 0.01 rather than 0.001 for the Kendall rank correlation coefficient test between each pair of dimensions' electrode weightings vectors. This means that we were more conservative in the sense that we were more likely to flag neural dimensions as non-orthogonal. For measuring the angle between the $CIS_1$ dimension and the first jPC plane (*Figure 4—figure supplement 1E*), we used the *subspacea* package for MATLAB, downloaded from https://www.mathworks.com/matlabcentral/fileexchange/55-subspacea-m (*Knyazev and Argentati, 2002*). To test whether the $CIS_1$ was significantly non-orthogonal to each of the jPCA dimensions individually, we used the same Kendall rank correlation test as described above.

## Rotatory dynamics

The second form of neural population structure we tested for was rotatory (i.e. oscillatory) low-dimensional dynamics. We applied methods previously developed to identify and quantify rotatory dynamics in motor cortex during NHP arm reaching (*Churchland et al., 2012*). These methods were also recently applied to show rotatory dynamics during hand movements of BrainGate2 study participants (*Pandarinath et al., 2015*). Churchland, Cunningham and colleagues introduced the jPCA dimensionality reduction technique for this purpose; we employed their MATLAB analysis package, downloaded from https://churchland.zuckermaninstitute.columbia.edu/content/code.

Trial-averaged firing rates for each word speaking condition were generated from 150 ms before to 100 ms after acoustic onset to capture an epoch when speech-producing articulator movements were being produced. Following *Churchland et al. (2012)* and *Pandarinath et al. (2015)*, these firing rates were soft-normalized with a 10 Hz offset and smoothed with a Gaussian kernel; we used a 30 ms s.d. kernel as in *Pandarinath et al. (2015)*. These firing rates were 'centered' by subtracting the across-condition mean firing rate of each unit at each time point, and then sampled every 10 ms. The dimensionality of these data was reduced via PCA to six; this ensured that rotatory dynamics

would be sought within population activity components that were strongly present in the data. jPCA was then used to find planes with rotatory structure within this six-dimensional subspace. The jPCs are found by fitting the following linear dynamical system:

$$\dot{\mathbf{x}} = \mathbf{M}_{skew}\, \mathbf{x} \qquad (2)$$

where $\mathbf{x}$ is the neural state (i.e. the PCA dimensionality-reduced population firing rate) at a given time, $\dot{\mathbf{x}}$ is its time derivative, and $\mathbf{M}_{skew}$ is constrained to be a skew-symmetric matrix. The first jPCA plane, which has the strongest rotatory dynamics, is defined by the two complex eigenvectors of $\mathbf{M}_{skew}$ with the largest eigenvalues. The choice of real vectors $jPC_1$ and $jPC_2$ within this plane is arbitrary and, following convention, were chosen such that conditions' activities are maximally spread along $jPC_1$ at the start of the analysis epoch. *Figure 5A* plots the trial-averaged population activity during speaking each word (after subtracting the across-conditions mean) in this top jPCA plane. The red/black/green color of each word condition's neural trajectory corresponds to its projection along $jPC_1$ at the start of the epoch; this display style is intended to assist in observing that amplitude and phase tend to unfold lawfully from the initial neural state. It is worth emphasizing that each jPC is simply a linear weighting of different units' firing rates, and that the six jPCs form an orthonormal basis set that spans the same subspace as the top six PCs. The strength of rotatory dynamics was quantified as the goodness of fit for *Equation 2* for a $2 \times 2$ $\mathbf{M}_{skew}$ in the first jPCA plane, and for a $6 \times 6$ $\mathbf{M}_{skew}$ in the 6-dimensional subspace defined by the top 6 PCs of the data. *Figure 5B* reports this 6D fit quality.

## Statistical testing of rotatory dynamics

To calculate the statistical significance of rotatory population dynamics structure in our data, we applied the 'neural population control' approach developed by Elsayed and Cunningham (*Elsayed and Cunningham, 2017*). This method was developed to address a potential concern that many specific phenomena that an experimenter could test for (such as fitting low-dimensional rotatory dynamics to neural data) can be found 'by chance' in a sufficiently high-dimensional, complex dataset such as the time-varying firing rates of many neurons. To address this, the method tests whether an observed feature of the population activity is 'novel' in the sense that it cannot be trivially predicted from known simpler features in the data. This is achieved by constructing surrogate datasets with simple population structure (in the form of means and correlations across time, neurons, and behavioral conditions) matched to the real data. If the neural recordings contain population-level structure that is coordinated above and beyond these first- and second-order features, then the quantification method used to describe this structure should return a stronger read-out when applied to the original dataset than to the surrogate datasets.

In our case, we used this approach to test whether it is 'surprising' to see rotatory dynamics in neural population data, given the particular smoothness across time, units, and word speaking conditions present in these data. A similar approach was used in *Elsayed and Cunningham (2017)* to further validate the original rotatory dynamics finding of *Churchland et al. (2012)*. We used the MATLAB code associated with *Elsayed and Cunningham (2017)* from https://github.com/gamaleldin/TME to generate 1000 surrogate datasets with time, neuron, and condition means and covariance matched to the real data using the tensor maximum entropy algorithm ('surrogate-TNC' flag in *fitMaxEntropy*). We then ran the same jPCA analyses described above on these surrogate datasets and recorded the rotation dynamics goodness of fit for the best $\mathbf{M}_{skew}$ matrix found for each surrogate dataset. This distribution of surrogate dataset $R^2$ values serves as a null distribution for significance testing: we calculated a p-value by counting how many of the surrogate datasets' $R^2$ exceeded that of the true original dataset.

## Neural state trajectory videos

The goal of *Videos 2* and *3* is to visualize how participant T5's neural population activity undergoes a condition-invariant 'kick' after the go cue (*Figure 4*) followed by rotatory dynamics around acoustic onset (*Figure 5*). To do so, we projected the ensemble neural activity during speaking short words into a lower dimensional neural state space designed to capture both the prominent condition-invariant component (hence, $CIS_1$ is one of the three projection dimensions) and rotatory dynamics (hence, the remaining two dimensions are the top jPCA plane). Plotting the word conditions' neural

state trajectories in the same state space required harmonizing the slightly different pre-processing used in the dPCA (*Figure 4*) and jPCA (*Figure 5*) analyses. Specifically, the trial-averaged neural trajectories in these videos were generated using the 30 ms s.d. Gaussian smoothing and 5 Hz soft-normalization parameters from the dPCA analysis. The $CIS_1$ dimension was found by applying dPCA to the same time epoch as in *Figure 4* (200 ms before go to 400 ms after go), and the $jPC_1$ and $jPC_2$ dimensions were found by applying jPCA to the same time epoch as in *Figure 5* (150 ms before to 100 ms after acoustic on).

To facilitate viewing the neural state trajectories in three (orthogonal) dimensions consisting of $[CIS_1, jPC_1, jPC_2]$, for these videos only we enforced that $jPC_1$ and $jPC_2$ be orthogonal to $CIS_1$ (empirically, without this constraint the top jPCA plane was 75° from the $CIS_1$, as shown in *Figure 4—figure supplement 1E*). To do so, prior to running jPCA, the trial-averaged firing rates were projected into the null space of the $CIS_1$ (the orthogonal complement of the first column of the encoder matrix returned by dPCA). That is, instead of jPCA operating on the $E = 96$ electrodes firing rates, it operated on a $96-1 = 95$ dimensional projection of the firing rates. The overall consequence of these decisions is that in these videos, the neural state is projected onto the exact same $CIS_1$ dimension as in *Figure 4*, whereas the $jPC_1$ and $jPC_2$ dimensions differ slightly from *Figure 5* due to the aforementioned spike train pre-processing differences and $CIS_1$ orthogonalization.

## Acknowledgements

We thank participants T5, T8, and their caregivers for their dedicated contributions to this research; Nancy Lam for administrative support; Dr. Sydney Cash and Dr. Laura Ball for helpful discussions; and Dr. Marc Slutzky for helpful discussions and providing the list of many words for sampling many distinct phonemes.

This work was supported by an ALS Association Milton Safenowitz Postdoctoral Fellowship, A. P. Giannini Foundation Postdoctoral Fellowship, Wu Tsai Neurosciences Institute Interdisciplinary Scholar Award, and Burroughs Wellcome Fund Career Award at the Scientific Interface (SDS); NSF Graduate Research Fellowship DGE – 1656518 and Regina Casper Stanford Graduate Fellowship (GHW); Larry and Pamela Garlick, Samuel and Betsy Reeves (KVS, JMH); NIDCD R01DC014034 (JMH); Office of Research and Development, Rehabilitation R and D Service, Department of Veterans Affairs N9288C, A2295R, B6453R, Executive Committee on Research of Massachusetts General Hospital, NIDCD R01DC009899 (LRH); NICHD R01HD077220 (RFK); NINDS 5U01NS098968-02 (LRH); Howard Hughes Medical Institute (KVS). The funders had no role in study design, data collection and interpretation, or the decision to submit the work for publication.

## Additional information

### Competing interests

Leigh R Hochberg: The MGH Translational Research Center has clinical research support agreements with Paradromics and Synchron Med, for which LRH provides consultative input. LRH is also a consultant for Neuralink. Krishna V Shenoy: is a consultant for Neuralink Corp and on the scientific advisory boards of CTRL-Labs Inc, MIND-X Inc, Inscopix Inc, and Heal Inc. Jaimie M Henderson: is a consultant for Neuralink Corp, Proteus Biomedical and Boston Scientific, and serves on the Medical Advisory Boards of Enspire DBS and Circuit Therapeutics. The other authors declare that no competing interests exist.

### Funding

| Funder | Grant reference number | Author |
| --- | --- | --- |
| ALS Association | Milton Safenowitz Postdoctoral Fellowship 17-PDF-364 | Sergey D Stavisky |
| A.P. Giannini Foundation | Postdoctoral Research Fellowship | Sergey D Stavisky |

| Wu Tsai Neurosciences Institute | Interdisciplinary Scholar Award | Sergey D Stavisky |
| --- | --- | --- |
| Burroughs Wellcome Fund | Career Award at the Scientific Interface | Sergey D Stavisky |
| National Science Foundation | Graduate Research Fellowships Program DGE - 1656518 | Guy H Wilson |
| Regina Casper Stanford Graduate Fellowship | DGE - 1656518 | Guy H Wilson |
| Eunice Kennedy Shriver National Institute of Child Health and Human Development | R01HD077220 | Robert F Kirsch |
| U.S. Department of Veterans Affairs | Office of Research and Development, Rehabilitation R&D Service N9228C | Leigh R Hochberg |
| U.S. Department of Veterans Affairs | Office of Research and Development, Rehabilitation R&D Service B6453R | Leigh R Hochberg |
| U.S. Department of Veterans Affairs | Office of Research and Development, Rehabilitation R&D Service A2295R | Leigh R Hochberg |
| U.S. Department of Veterans Affairs | Office of Research and Development, Rehabilitation R&D Service N2864C | Leigh R Hochberg |
| National Institute on Deafness and Other Communication Disorders | R01DC009899 | Leigh R Hochberg |
| Executive Committee on Research of Massachusetts General Hospital | | Leigh R Hochberg |
| National Institute of Neurological Disorders and Stroke | 5U01NS098968-02 | Leigh R Hochberg |
| Larry and Pamela Garlick | | Krishna V Shenoy Jaimie M Henderson |
| Samuel and Betsy Reeves | | Krishna V Shenoy Jaimie M Henderson |
| National Institute on Deafness and Other Communication Disorders | R01DC014034 | Jaimie M Henderson |
| Howard Hughes Medical Institute | | Krishna V Shenoy |

The funders had no role in study design, data collection and interpretation, or the decision to submit the work for publication.

## Author contributions

Sergey D Stavisky, Conceptualization, Data curation, Software, Formal analysis, Funding acquisition, Investigation, Visualization, Methodology; Francis R Willett, Software, Formal analysis; Guy H Wilson, Formal analysis, Visualization; Brian A Murphy, Paymon Rezaii, Donald T Avansino, Jonathan P Miller, Investigation; William D Memberg, Investigation, Project administration; Robert F Kirsch, Leigh R Hochberg, Supervision, Funding acquisition, Project administration; A Bolu Ajiboye, Supervision, Funding acquisition; Shaul Druckmann, Formal analysis, Supervision; Krishna V Shenoy, Jaimie M Henderson, Conceptualization, Supervision, Funding acquisition, Project administration

## Author ORCIDs

Sergey D Stavisky ID https://orcid.org/0000-0002-5238-0573
Guy H Wilson ID https://orcid.org/0000-0003-0961-1994
Paymon Rezaii ID https://orcid.org/0000-0002-4803-0853
Leigh R Hochberg ID https://orcid.org/0000-0003-0261-2273
A Bolu Ajiboye ID https://orcid.org/0000-0002-9402-1165
Jaimie M Henderson ID https://orcid.org/0000-0002-3276-2267

## Ethics

Clinical trial registration NCT00912041.

Human subjects: The two participants in this study were enrolled in the BrainGate2 Neural Interface System pilot clinical trial (ClinicalTrials.gov Identifier: NCT00912041). The overall purpose of the study is to obtain preliminary safety information and demonstrate proof of principle that an intracortical brain-computer interface can enable people with tetraplegia to communicate and control external devices. Permission for the study was granted by the U.S. Food and Drug Administration under an Investigational Device Exemption (Caution: Investigational device. Limited by federal law to investigational use). The study was also approved by the Institutional Review Boards of Stanford University Medical Center (protocol #20804), Brown University (#0809992560), University Hospitals of Cleveland Medical Center (#04-12-17), Partners HealthCare and Massachusetts General Hospital (#2011P001036), and the Providence VA Medical Center (#2011-009). Both participants gave informed consent to the study and publications resulting from the research, including consent to publish photographs and audiovisual recordings of them.

## Decision letter and Author response

Decision letter https://doi.org/10.7554/eLife.46015.sa1
Author response https://doi.org/10.7554/eLife.46015.sa2

# Additional files

## Supplementary files

- Source data 1. Breathing data.
- Source data 2. Classification data.
- Source data 3. Dynamics data.
- Source data 4. PSTHs sorted units data.
- Source data 5. Syllables PSTHS TCs data.
- Source data 6. Tuning and behavior data.
- Source data 7. Video go aligned data.
- Source data 8. Video and dynamics speak aligned data.
- Source data 9. Words PSTHs TCs data.
- Transparent reporting form

## Data availability

The sharing of the raw human neural data is restricted due to the potential sensitivity of this data. These data are available upon request to the senior authors (KVS or JMH). To respect the participants' expectation of privacy, a legal agreement between the researcher's institution and the BrainGate consortium would need to be set up to facilitate the sharing of these datasets. Processed data is provided as source data, and analysis code is available at https://github.com/sstavisk/speech_in_dorsal_motor_cortex_eLife_2019 (copy archived at https://github.com/elifesciences-publications/speech_in_dorsal_motor_cortex_eLife_2019).

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
