## [Decision Letter]

**Acceptance summary:**

The paper by Stavisky et al. provides important innovation by providing a first account of the population dynamics relating to motor control of a body part within a 'canonical' motor area of another body part. The authors utilise electrode arrays implanted in participants with paralysis to identify mouth and speech-related motor processing in the hand/arm area of motor cortex. They analyse single neurons and neural population activity during speech, and demonstrate selective responses for spoken words, syllables and orofacial movements, resulting in high classification accuracy across words/syllables. The authors further interrogate the population dynamics, previously established for hand and arm movements in this cortical area, in search for a common neural dynamics underlying motor control from this area. The paper stood out in its quality and rigour of data analysis and clarity of conceptualisation. As such, the paper offers potential innovation on multiple fronts, from basic principles of brain organisation for motor control to assistive technologies via brain-machine interfaces, and is expected to appeal to a broad audience across multiple sub-fields.

**Decision letter after peer review:**

Thank you for submitting your article "Neural ensemble dynamics in dorsal motor cortex during speech in people with paralysis" for consideration by *eLife*. Your article has been reviewed by three peer reviewers, including Tamar R Makin as the Reviewing Editor and Reviewer #1, and the evaluation has been overseen by Barbara Shinn-Cunningham as the Senior Editor. The following individuals involved in review of your submission have agreed to reveal their identity: Juan Álvaro Gallego (Reviewer #2), and Sophie K Scott (Reviewer #3).

The reviewers have discussed the reviews with one another and the Reviewing Editor has drafted this decision to help you prepare a revised submission.

The paper by Stavisky et al. utilises electrode arrays implanted in participants with paralysis to identify mouth and speech-related motor processing in the hand/arm area of motor cortex. The authors analyse single neurons and neural population activity during speech, and demonstrate selective responses for spoken words, syllables and orofacial movements, resulting in high classification accuracy across words/syllables. The authors further interrogate the population dynamics, previously established for hand and arm movements in this cortical area, in search for a common neural dynamics underlying motor control from this area. While the observation that body-part assignment along the motor 'homunculus' is more broadly distributed than commonly regarded is not in itself new (as highlighted by Penfield in his seminal work), the current study provides important innovation by providing a first account of the population dynamics relating to a 'misplaced' body part. Overall, the paper is well-written, the key analyses are sound, and the results highly interesting. However, the reviewers agreed that the characterisation of the neural results in the specific context of speech production could be improved by taking into consideration the wide range different patterns of motor control, from breath control, laryngeal engagement and control of the articulators. Conceptually, the reviewers felt that further discussion is required in order to place the findings in context, with respect to face-hand overlap, as elaborated below.

Essential revisions:

1) There seems to a be a missed opportunity to consider the varying motor demands for the larynx/tongue/mouth/lips across stimuli, or even laryngeal and speech breathing mechanisms (note that metabolic breathing, which the authors have accounted for, is entirely different). In the current analysis, each of the syllables/words is studies in isolation from the others, but in term of motor control there should be some clear similarities and distinctions across these stimuli, which could also be further linked with the motor demands of the orofacial movements. For example, decoding accuracy might vary depending on similarities for motor control of these various motor mechanisms involved in speech production. This will go a long way showing that the findings observed here relate to the motor processing relating to articulation, rather than other forms of information content. More generally, these important considerations relating to the mechanisms of speech productions need to be more thoughtfully integrated in the manuscript, the authors might like to consult with an expert for this purpose.

2) Conceptually, there is a need to better consider why facial information exists in the hand area. Is that because of a unique association between the mouth and hand for language? Here the authors might like to consider commonality of gestures, and consider whether this a semantic or timing-based gestural relationship, or both? Another interesting link to consider is between speaking and reading/writing? Or topographic proximity? Alternatively, could there be nothing special between the hand and the face – there could also exist information in the hand area for feet movements? Related to that, for participant T8 – the electrode arrays are too dorsomedial to be considered as the hand area. So it seems that the results suggest that orofacial/speed-related information is present throughout motor cortex. This brings us back to the question whether the SCI, and expected E/I balance changes in the deafferented cortex might play a role in the present findings. The reviewers agreed that the conceptual framework of the study could benefit from further justification/interpretation.

3) The results are often reported in descriptive terms but are not statistically tested, making it difficult to accept some of the characteristics offered by the authors. The reviewers would like to see more quantifications in the paper, including: percentage variance explained as function of the number of components (for PCA and dPCA), pairwise angles between CI and CD dPCs together with their significance threshold (Kobak et al. proposed a method on their paper), etc. Moreover, couldn't the authors apply these methods to the syllables datasets even if they had less trials, they were sorter, and the neural activity was less consistent (they can compensate for this with the speech I think)?

4) While the classification accuracy is impressive, it’s important to dissociate between the motor control component to others relating to perception and intention. The authors mention that responses during the audio prompt were small and thus they couldn't disambiguate whether they reflect perception, movement preparation, etc (subsection “Speech-related activity in dorsal motor cortex”, first paragraph). Based on Video 1, it seems to one reviewer that there's some modulation during the prompts. Is it possible to classify rapid responses in a small window centered around the auditory cue? If decoding accuracy is significantly greater during articulation, it might be provide support for the overall interpretation of the findings.

5) Similarly, can the authors explore whether there are any rotation motifs around the prompt? This would help answer the question whether this is an inherent network property of the area, or whether it is specific for movement planning.

6) The neural population analyses look quite different for the two patients: 1) for T8 there's only one CI dPC, and it explains roughly the same amount of variance as the leading CD dPC, whereas for T5 there are two CI dPCs that explain several times more variance than any CD dPC; 2) the rotational structure identified with jPC is not above the chance level for T8, only for T5. We understand that these differences may very well be motivated by the worse quality of T8's arrays, but the authors should be more cautious in some parts of the paper given these differences and their n=2, e.g., in the Abstract. Moreover, this difference should be addressed to a greater extent in the Discussion.

7) The authors suggest that the hand area might play a role in speech production. Here they seem to conflate correlation with causation – their findings do not provide any support that this decoding information available in the hand area is actually utilised during speech motor control.

---

## [Author Response]

[…] Overall, the paper is well-written, the key analyses are sound, and the results highly interesting. However, the reviewers agreed that the characterisation of the neural results in the specific context of speech production could be improved by taking into consideration the wide range different patterns of motor control, from breath control, laryngeal engagement and control of the articulators. Conceptually, the reviewers felt that further discussion is required in order to place the findings in context, with respect to face-hand overlap, as elaborated below.

In response to the reviewers’ feedback, we have added additional data and analyses to better relate this activity to the motoric demands of speech and volitional breath control, and we have expanded the Discussion to better place these findings in context. We have also added additional discussion of interpretation limitations based on the reviewers’ feedback. These changes, as well as many others, are described in more detail in responses to specific comments below.

With regards to the novelty of a broadly distributed homunculus, we appreciate the feedback that this is not entirely new. We were surprised by our results because we are unaware of previous studies showing this extent of distribution (i.e., face activity in this hand knob area), even after taking into account recent work describing fractured somatotopy within major body regions and the partially overlapping distributions of the original Penfield work (which was both due to within-subject and across-subjects effects). Thus, on this topic we hope (and believe) that our report of mixed tuning at the level of single neurons is an interesting and novel contribution that provides additional data of relevance to an emerging view of broad motor maps.

Essential revisions:1) There seems to a be a missed opportunity to consider the varying motor demands for the larynx/tongue/mouth/lips across stimuli, or even laryngeal and speech breathing mechanisms (note that metabolic breathing, which the authors have accounted for, is entirely different). In the current analysis, each of the syllables/words is studies in isolation from the others, but in term of motor control there should be some clear similarities and distinctions across these stimuli, which could also be further linked with the motor demands of the orofacial movements. For example, decoding accuracy might vary depending on similarities for motor control of these various motor mechanisms involved in speech production. This will go a long way showing that the findings observed here relate to the motor processing relating to articulation, rather than other forms of information content. More generally, these important considerations relating to the mechanisms of speech productions need to be more thoughtfully integrated in the manuscript, the authors might like to consult with an expert for this purpose.

Thank you for your feedback that the manuscript could, and should, do more to relate the observed speech-related neural activity with the underlying motor actions (the movements of the larynx/tongue/mouth/lips, and breathing) required to produce these sounds. We absolutely agree. We describe below (1) new measurements and analyses that we plan to do (but consider to be future research) now that these new cortical responses are known about and initially characterized, and (2) the considerable amount of new analysis work, *including newly collected data*, that we have done and include in the revised manuscript to directly address these points.

Regarding (1) above, we start out below by first describing – in some detail – for the reviewers and editors our future research roadmap which will invest considerably in pursuing exactly these questions. These are major research endeavors in and of themselves which will span considerable periods of time (likely a year or more, involving new FDA approval and new research infrastructure), and will lead to subsequent full reports. Regarding (2) above, we then turn to a considerable amount of new data analysis that we have done, including the use of newly collected data, that we have integrated into the revised manuscript (including a new figure), to directly address the questions raised. We believe that these analyses and results add considerably to the manuscript and, again, we are grateful for the reviewers’ and editors’ suggestion to more deeply pursue this topic.

(1) We think the best way forward toward answering in full detail the question regarding neural mechanisms of speech production is to collect new data and bring in new technical capabilities so that we can either measure speech articulator movements directly, or infer them from recorded audio (for example using new audio-articulator inversion ‘AAI’ methods like in Chartier et al., 2018, which unfortunately are not yet publicly available). Such data would overcome our current limitations: although we know what syllable/word the participants spoke and what prompted orofacial movements they made, we don’t have moment-by-moment kinematic measurements. Also, our original data sampled relatively few orofacial and speech movements. Without the underlying kinematic measurements and without comprehensive sampling of different combinations of kinematics, it becomes very difficult to attribute measured neural activity to specific articulatory degrees-of-freedom, since even short syllables or prompted movements are coordinated high-dimensional movements of multiple articulators.

We are investing considerably to be able to collect data that overcomes this limitation in the future, with: (A) a major purchase of Electromagnetic Midsagittal Articulography equipment, (B) planning for making a request for FDA regulatory approval to use these (uncomfortable but safe) techniques with our clinical trial participants, if they agree to do so, and (C) a newly established collaboration. However, doing all of this well this is a major undertaking – that will lead to one or more major additional full reports – and that realistically will extend out for over a year. Thus we view this as future work, and outside the scope of the present manuscript. We have added a Discussion paragraph (see below) laying out that we think this is a promising future direction in which to build on the present work. Our hope is that the present work, which provides a first description of this speech/orofacial single neuron activity in this cortical area, will lay the groundwork for this subsequent work.

This is a key way in which we believe we are addressing, as requested, the reviewer’s comment that “these important considerations relating to the mechanisms of speech productions need to be more thoughtfully integrated in the manuscript” in the longer term. Now we turn to (2), which addresses this request more directly by describing new analyses with new data that we have performed and added to the revised manuscript.

(2) We recognize that the manuscript would benefit from additional examination of how the neural correlates of different spoken sounds relate to their motor demands, and we are grateful to the reviewers and editors for suggesting this. To this end, we have performed a key new analysis on newly collected data. We asked participant T5 to speak 420 different words (3 times each) chosen to broadly sample 41 American English phonemes (we unfortunately could not repeat this new data collection in participant T8, who is no longer in the clinical trial). We then hand-segmented each of these words’ audio data into individual phonemes, and compared the neural ensemble correlates of these phonemes. This new result reveals that phonemes’ neural correlates clustered based on phonetic groupings and place of articulation, consistent with this activity being related to the underlying motor demands for producing the phonemes. See Figure 1—figure supplement 5.

We view this new result as consistent with the hypothesis that this neural activity is related to speech articulator movements. We believe that this publication helps motivate future work, by our group and potentially by other groups too, to further pursue the relationship between neural activity and articulatory kinematics.

This new dataset and its task are described in the ‘Many words task’ Materials and methods section; further details of the analysis are described in the ‘Comparing different phonemes’ neural correlates’ Materials and methods section. The added Results section text reads:

“Second, analysis of an additional dataset in which participant T5 spoke 41 different phonemes revealed that neural population activity showed phonemic structure (Figure 1—figure supplement 5): for example, when phonemes were grouped by place of articulation (Bouchard et al., 2013; Lotte et al., 2015; Moses et al., 2019), population firing rate vectors were significantly more similar between phonemes within the same group than between phonemes in different groups (p<0.001, shuffle test).”

Finally, regarding this figure, we would be happy to elevate it to a figure in the main text (instead of a supplementary figure) if the reviewers and/or editors recommend this. While we indeed view this as an important analysis and figure, and are grateful that the reviewers and editors suggested that we pursue this, we also want to be mindful of not making the manuscript too long or over-emphasizing a single participant result. Thus, we pose this question here.

We share your prediction that breathing may contribute a component of our observed speech-related neural activity, and that it will be useful to study the neural correlates of breathing in the context of speech production, which is distinct from the metabolic breathing we studied here. As discussed above, we think that the best way forward on this question is to include breathing amongst many continuous speech articulatory kinematics in a future study, so that each of these movements’ distinct partial correlations with neural activity can be disentangled.

With that said, we are grateful to the reviewers and editors for suggesting that we pursue this a bit further, as we believe that we were able to take an additional step towards understanding breath-related activity. We did so by analyzing data from an additional ‘instructed breathing task’ in which breathing was under the participant’s conscious control. This new behavioral context is now analyzed alongside the previously presented unattended breathing data. The updated Figure 2—figure supplement 2, shows that volitional breathing also modulates hand knob cortex.

Please note that compared to the initial submission’s Figure 2—figure supplement 2, the shuffle distributions (panel C, D horizontal dashed lines) have shifted; this is because a) we caught and fixed a bug in the shuffle ordering code, and b) we changed the significance threshold to 0.01 (from 0.001) to maintain sensitivity after this fix and accommodate the reduced trial count of the new instructed breathing condition. This change does not affect our conclusions: besides moving the panel C, D lines, the net effect is that we now report that 17 (rather than 18) of the single neurons were significantly correlated with breathing. We apologize for this mistake and have carefully checked our code throughout the manuscript.

Below we have copied the updated Discussion paragraph that summarizes our evidence supporting that the observed neural activity is related to motor control, and suggests future work looking at speech kinematics:

“Our data suggest that the observed neural activity reflects movements of the speech articulators (the tongue, lips, jaw, and larynx): modulation was greater during speaking than after hearing the prompt; the same neural population modulated during non-speech orofacial movements; and in T5, the neural correlates of producing different phonemes grouped according to these phonemes’ place of articulation. […] A deeper understanding of how motor cortical spiking activity relates to complex speaking behavior will require future work connecting it to continuous articulatory (Chartier et al., 2018; Conant et al., 2018; Mugler et al., 2018) and respiratory kinematics and, ideally, the underlying muscle activations.”

2) Conceptually, there is a need to better consider why facial information exists in the hand area. Is that because of a unique association between the mouth and hand for language? Here the authors might like to consider commonality of gestures, and consider whether this a semantic or timing-based gestural relationship, or both? Another interesting link to consider is between speaking and reading/writing? Or topographic proximity? Alternatively, could there be nothing special between the hand and the face – there could also exist information in the hand area for feet movements? Related to that, for participant T8 – the electrode arrays are too dorsomedial to be considered as the hand area. So it seems that the results suggest that orofacial/speed-related information is present throughout motor cortex. This brings us back to the question whether the SCI, and expected E/I balance changes in the deafferented cortex might play a role in the present findings. The reviewers agreed that the conceptual framework of the study could benefit from further justification/interpretation.

Thank you, and we agree that the manuscript would benefit from more discussion of why there might be face information in “hand” area of motor cortex. While we originally speculated that this was due to the kinds of hand-mouth linkages enumerated by the reviewers and editors (based on previous studies such as those referenced in our Introduction), new results from our group (currently in the pre-print stage) have made us re-evaluate this interpretation. Inspired by finding face activity in hand knob area, we then tested whether there was modulation during actual and attempted movements of other body parts, including the neck, ipsilateral arm, and legs, exactly as you proposed. We found that indeed there is representation of every body part tested (Willett, et al., bioRxiv 2019). In light of this, our interpretation of these results is that finding speech-related activity in this cortical area is a consequence of motor representations being much more distributed at the single-neuron level than we previously imagined, rather than a “special” hand-face relationship (though we can’t rule that out, and it would be interesting to explicitly examine coordinated hand-face movements in future work). We have updated our Discussion accordingly, and we have also added a new paragraph that explicitly calls out that we are far from resolving what the “purpose” of this speech-related activity in hand knob area is (if any) and that we feel this is an important, though difficult, question for future research:

“There are three main findings from this study. […] Thus, the observed neural overlap between hand and speech articulators may be a consequence of distributed whole-body coding, rather than a privileged speech-manual linkage.”

“Assuming that these results are not due to injury-related remapping, we are left with the question of *why* this speech-related activity is found in dorsal “arm and hand” motor cortex. […] We anticipate that it will require substantial future work to understand why speech-related activity co-occurs in the same motor cortical area as arm and hand movement activity, but that this line of inquiry may reveal important principles of how sensorimotor control is distributed across the brain (Musall et al., 2019; Stringer et al., 2019).”

It is our hope that an impact of this manuscript will be to help motivate further work to understand this (fascinating, to us) phenomenon and better appreciate the complexity of human motor representations.

We have expanded the Discussion paragraph about why we think the presence of speech activity in hand knob cortex is not due to cortical remapping following SCI to incorporate this new whole-body tuning evidence. We have also added new references. This paragraph is reproduced below for convenience:

“An important unanswered question, however, is to what extent these results were potentially influenced by cortical remapping due to tetraplegia. […] While these threads of evidence argue against remapping, definitively resolving this ambiguity would require intracortical recording from this eloquent brain area in able-bodied people.”

Regarding the placement of T8’s arrays: placement was guided anatomically by definitively identifying Yousry’s “hand knob” area, which has distinctive contours on volumetrically obtained MRI images and can be identified with a high degree of certainty (>97%) (described in Yousry, 1997). That said, we recognize that the extent of across-individual anatomy differences could raise questions about the accuracy and utility of generalized terms like “hand knob”, despite its adoption by neurosurgeons as a distinct anatomical structure. As further evidence for correct array placement, the functional properties of this area (strong hand and arm-related tuning) are also consistent with these arrays being in the same hand area as T5’s. We have added additional details to the Materials and methods to explain how the arrays were targeted:

“Both participants had two 96-electrode Utah arrays (1.5 mm electrode length, Blackrock Microsystems, USA) neurosurgically placed in dorsal ‘hand knob’ area of the left (motor dominant) hemisphere’s motor cortex. Surgical targeting was stereotactically guided based on prior functional and structural imaging (Yousry, 1997), and subsequently confirmed by review of intra-operative photographs.”

3) The results are often reported in descriptive terms but are not statistically tested, making it difficult to accept some of the characteristics offered by the authors. The reviewers would like to see more quantifications in the paper, including: percentage variance explained as function of the number of components (for PCA and dPCA), pairwise angles between CI and CD dPCs together with their significance threshold (Kobak et al. proposed a method on their paper), etc. Moreover, couldn't the authors apply these methods to the syllables datasets even if they had less trials, they were sorter, and the neural activity was less consistent (they can compensate for this with the speech I think)?

Thank you for pointing out that our neural population dynamics results and claims will be more strongly supported with additional quantifications and the inclusion of the syllables datasets. We have generated a new Figure 4—figure supplement 1 that provides these additional details, including cumulative variance explained for dPCA and jPCA and pairwise angles between dPCs (including the significance testing from Kobak et al., 2016). These quantifications are also now described in the ‘Condition-invariant signal’ Materials and methods section.

As per your suggestion, this supplementary figure also includes each participant’s syllables datasets, as well as two new T5 replication datasets. These additional ‘T5-5words-A’ and ‘T5-5words-B’ datasets were collected as part of a follow-up study, but we are happy to pull them in to this work and process them using the exact same analysis parameters used for the original datasets to build more confidence in the robustness of our findings. We believe that the manuscript is substantially strengthened by showing the consistency of these two neural population dynamics motifs across more datasets.

The additional datasets are discussed in the following updated Results passages:

“We found that these two prominent population dynamics motifs were indeed also present during speaking. […] These results were also robust across different choices of how many dPCs to summarize the neural population activity with (Figure 4—figure supplement 2).”

“Lastly, we looked for rotatory population dynamics around the time of acoustic onset. Figure 5A shows ensemble firing rates projected into the top jPCA plane. […] As was the case for the condition-invariant dynamics, these results were also consistent across additional datasets (Figure 4—figure supplement 1E-H) and across the choice of how many PCA dimensions in which to look for rotatory dynamics (Figure 4—figure supplement 2B).”

Please note that the Figure 4—figure supplement 1C dPC pairwise angles insets are a superset of the information provided in the original Figure 4 CIS_1_ vs. CD_1,2_ insets, so we have removed those. We now use a similar visual format in the new Figure 4—figure supplement 1E to compare the CIS_1_ versus the top jPCA plane, which we think is an interesting comparison to document. Perhaps unsurprisingly, the CIS_1_ is nearly orthogonal to the jPC dimensions, though we are careful to note that this need not be the case and that the model of a CIS that shifts dynamics into a different regime for movement generation does not require this to be the case (one could even imagine a CIS that shifts the neural state to a very different position in the exact same neural subspace also acting as a “trigger” for rotatory dynamics). The end of the Results now reads:

“We note that existing models of how a condition-invariant signal “kicks” dynamics into a different state space region where rotatory dynamics unfold (Kaufman et al., 2016; Sussillo et al., 2015) do not require that the CIS and rotatory dynamics must be orthogonal, but in these data we did observed that the CIS_1_ and jPCA dimensions were largely orthogonal (Figure 4—figure supplement 1E).”

4) While the classification accuracy is impressive, it’s important to dissociate between the motor control component to others relating to perception and intention. The authors mention that responses during the audio prompt were small and thus they couldn't disambiguate whether they reflect perception, movement preparation, etc (subsection “Speech-related activity in dorsal motor cortex”, first paragraph). Based on Video 1, it seems to one reviewer that there's some modulation during the prompts. Is it possible to classify rapid responses in a small window centered around the auditory cue? If decoding accuracy is significantly greater during articulation, it might be provide support for the overall interpretation of the findings.

There is indeed (small) modulation after the prompt, which by the way we now quantify in a better way. Thank you for suggesting that we further quantify how much word/syllable-specific information is present in this prompt activity using a similar decoding approach as when decoding the speaking epoch activity. We have added this analysis, which shows very poor prompt-epoch classification performance, to the manuscript (see Figure 3C). As you said, this further supports that this activity is related to speech production.

The updated Results passage is:

“We next performed a decoding analysis to quantify how much information about the spoken syllable or word was present in the time-varying neural activity. […] The much higher neural discriminability of syllables and words during speaking rather than after hearing the audio prompt is consistent with the previously enumerated evidence that modulation in this cortical area is related to speech production.”

5) Similarly, can the authors explore whether there are any rotation motifs around the prompt? This would help answer the question whether this is an inherent network property of the area, or whether it is specific for movement planning.

Thank you for this valuable suggestion. We have now performed the same jPCA rotatory dynamics analysis on an epoch (of the same length as the main Figure 5 analyses) shortly after the prompt. These results are shown in the newly added Figure 4—figure supplement 1H, and reveal that there were not rotatory dynamics after the prompt. In the interest of space, and since these prompt rotations were not significant, for this analysis we only show the variance explained summary statistic (right below the significant speech-epoch statistics, for contrast) and not the neural trajectories in the top jPCA plane. In addition to being relevant to the wider question of how ubiquitous (across behaviors) and specific (across time epochs) neural rotations are, this new analysis also provides an empirical control that jPCA doesn’t just trivially find significant rotations in any neural data.

These new results are described in the Results:

“Lastly, we looked for rotatory population dynamics around the time of acoustic onset. Figure 5A shows ensemble firing rates projected into the top jPCA plane. […] As was the case for the condition-invariant dynamics, these results were also consistent across additional datasets (Figure 4—figure supplement 1E-H) and across the choice of how many PCA dimensions in which to look for rotatory dynamics (Figure 4—figure supplement 2B).”

6) The neural population analyses look quite different for the two patients: 1) for T8 there's only one CI dPC, and it explains roughly the same amount of variance as the leading CD dPC, whereas for T5 there are two CI dPCs that explain several times more variance than any CD dPC; 2) the rotational structure identified with jPC is not above the chance level for T8, only for T5. We understand that these differences may very well be motivated by the worse quality of T8's arrays, but the authors should be more cautious in some parts of the paper given these differences and their n=2, e.g., in the Abstract. Moreover, this difference should be addressed to a greater extent in the Discussion.

Thank you for your feedback that there was not enough discussion of the neural population analyses differences between the two participants, and that these differences warrant caution when interpreting the results. We have made a number of manuscript changes which we believe address this:

First of all, after improving our analysis methods for quantifying population-wide task-related modulation, we realized that our speech initiation analysis epoch of 200 ms before the go cue to 400 ms after go, which we had originally selected when initially analyzing T5’s data, was a poor choice for participant T8 because his recorded neural modulation occurs later than T5’s. This choice of a premature (minimally modulating) epoch exacerbated the differences between participants (in addition to the worse array quality, as mentioned by the reviewer). We have now changed T8’s CIS analysis epoch to 100 ms to 700 ms after go, which re-focuses this analysis on a post-go “modulation ramp up” epoch that is more similar to T5’s. This yields CIS results that look much more similar between the two participants (see Figure 4).

We also described the reasoning for the different epochs in the Materials and methods:

“Trial-averaged firing rates were calculated from a speech initiation epoch of 200 ms before go cue to 400 ms after the go cue for T5, and 100 ms to 700 ms after the go cue for T8. T8’s epoch was shifted later relative to T5’s to account for T8’s later neural population activity divergence from the silent condition (Figure 1—figure supplement 4B).”

Second, we have changed the Results section presenting these results to address the differences between the two participants’ CIS results:

“We found that these two prominent population dynamics motifs were indeed also present during speaking.[…] This lower signal-to-noise ratio can also be appreciated in how the “elbow” of T8’s cumulative neural variance explained by PCA or dPCA components (Figure 4—figure supplement 1A, B) occurs after fewer components and explains far less overall variance.”

Third, we now also revisit the non-significant T8 rotatory dynamics result in the updated Discussion section, which now reads:

“Our third main finding is that two motor cortical population dynamical motifs present during arm movements were also significant features of speech activity. We observed a large condition-invariant change at movement initiation in both participants, and rotatory dynamics during movement generation in the one of two participants whose arrays recorded substantially more modulation.”

Fourth, we have added the n=2 to the Abstract:

“Speaking is a sensorimotor behavior whose neural basis is difficult to study with single neuron resolution due to the scarcity of human intracortical measurements. We used electrode arrays to record from the motor cortex ‘hand knob’ in two people with tetraplegia, an area not previously implicated in speech.”

Relatedly, we also now address the differences in the two participant’s decoding performance in the Discussion:

“That said, these results are only a first step in establishing the feasibility of speech BCIs using intracortical electrode arrays. […] We also observed worse decoding performance in participant T8, highlighting the need for future studies in additional participants to sample the distribution of how much speech-related neural modulation can be expected, and what speech BCI performance these signals can support.”

Also, please note that T5’s CIS dPCA Figure 4 plots have changed very slightly from the original submission because when revisiting these analyses, we noticed that we had been insufficiently regularizing the dimensionality reduction/variance partition process such that the dPCs didn’t generalize as well to held out data. We have now used the dPCA code’s built-in cross-validated regularization parameter optimization and verified that the resulting dimensionality reduction generalizes well if we do dPCA on only half the data and then compare the resulting dimensions and variance partitions when projecting the other (held-out) half of the data into these dPCs. We have added this detail to the ‘Condition-invariant signal’ Materials and methods section:

“Default *dpca* function parameters were used, with parameters numRep = 10 (repetitions for regularization cross-validation) and simultaneous = true (indicating that the single-trial neural data were simultaneously recorded across electrodes) for the *dpca_optimizeLambda* and *dpca_getNoiseCovariance* functions.”

7) The authors suggest that the hand area might play a role in speech production. Here they seem to conflate correlation with causation – their findings do not provide any support that this decoding information available in the hand area is actually utilised during speech motor control.

Thank you for pointing out that as originally written, our Discussion came across as suggesting that these data indicate a causal role of hand area in speech production. We apologize for this, as we absolutely agree that we have only observed correlation with speaking, and no evidence for a causal role. We have updated several sections of the Discussion, reproduced below, to be more cautious in speculating, what, if any, role this activity might have in speech or coordinating speech and hand movements:

“There are three main findings from this study. First, these data suggest that ‘hand knob’ motor cortex, an area not previously known to be active during speaking (Breshears et al., 2015; Dichter et al., 2018; Leuthardt et al., 2011; Lotte et al., 2015), may in fact participate, or at least receive correlates of, neural computations underlying speech production.”

“Assuming that these results are not due to injury-related remapping, we are left with the question of *why* this speech-related activity is found in dorsal “arm and hand” motor cortex. […] We anticipate that it will take substantial future work to understand why speech-related activity co-occurs in the same motor cortical area as arm and hand movement activity, but that this line of inquiry may reveal important principles of how sensorimotor control is distributed across the brain (Musall et al., 2019; Stringer et al., 2019).”